# *In vitro* and *in vivo* activity of sodium houttuyfonate and sodium new houttuyfonate against *Candida auris* infection by affecting adhesion, aggregation, and biofilm formation abilities

Guangyuan Yang,[1] Ruotong Yang,[1] Xiaoxiao Zhu,[1] Qianwen Xu,[1] Xiaojia Niu,[1,2] Chengui Miao,[1] Wenfan Wei,[1,2] Changzhong Wang,[1,2] Tianming Wang,[2] Daqiang Wu[1,2]

**ABSTRACT** *Candida auris* is a rapidly spreading multidrug-resistant fungus that causes fatal infections under certain global conditions. Sodium houttuyfonate (SH) and sodium new houttuyfonate (SNH) are stable derivatives of houttuynin (decyl aldehyde) extracted from *Houttuynia cordata*, both possessing antifungal and antibacterial pharmacological activities. However, the inhibitory effects of SH and SNH on *C. auris* remain unclear. Therefore, this study aims to evaluate the potential activity and possible mechanisms of SH and SNH as antifungal agents against *C. auris*. First, our results showed that SH and SNH exhibit significantly inhibitory activity against fluconazole-resistant *C. auris* strains, but do not possess effective fungicidal activity. In addition, transcriptome and RT-qPCR studies revealed that SH and SNH can repress the expression of genes related to adhesion, aggregation, and biofilm formation. Next, we observed that SH and SNH can disrupt the adhesion and aggregation of early-stage *C. auris*. Furthermore, using the XTT assay, crystal violet staining, and confocal laser scanning microscopy, we found that the biofilm formation ability of *C. auris* was disrupted by SH and SNH. We also found that SH and SNH can potentially increase chitin content and expose β-1,3-glucan in the cell wall. Finally, infection models using *Galleria mellonella* larvae and mice with systemic candidiasis demonstrated that SH and SNH significantly inhibited the colonization and pathological damage of *C. auris in vivo*. Therefore, our presented results suggest that SH and SNH can effectively inhibit the growth, adhesion, aggregation, and biofilm formation to treat its colonization and pathological damage to the host of *C. auris*.

**IMPORTANCE** Recently, the annual proportion of non-*C. albicans* infections has been rising. The most notable characteristic of *C. auris* is its resistance to drugs, including multidrug resistance, which results in treatment failures and poses significant challenges in controlling its spread. Sodium houttuyfonate (SH) and sodium new houttuyfonate (SNH) are effective and stable derivatives of houttuynin (decyl aldehyde) extracted from traditional Chinese herbal medicine Houttuynia cordata, both possessing antifungal and antibacterial pharmacological activities. However, the inhibitory effects of SH and SNH on *C. auris* remain unclear. Through *in vitro* and *in vivo* approaches, we have demonstrated that SH and SNH can effectively inhibit the growth, adhesion, aggregation, and biofilm formation to treat its colonization and pathological damage to the host of *C. auris*. Thus, our findings provide new insights into possible options for clinical applications in the anti-*C. auris*.

**KEYWORDS** *Candida auris*, sodium houttuyfonate, sodium new houttuyfonate, adhesion, aggregation, biofilm, virulence

Address correspondence to Daqiang Wu, daqwu@ahtcm.edu.cn, or Tianming Wang, wtm@ahtcm.edu.cn.

Guangyuan Yang and Ruotong Yang contributed equally to this article. Author order was determined based on the contributions.

The authors declare no conflict of interest.

See the funding table on p. 24.

The incidence and prevalence of invasive fungal infections have increased in recent years, with over 90% of these infections caused by *Candida albicans*, *Candida glabrata*, *Candida parapsilosis*, *Candida dubliniensis*, *Candida tropicalis*, and *Saccharomyces kudriavzevii* (formerly known as *Candida krusei*) (1–3). However, recently, *C. auris* has emerged as a serious global health threat due to its multidrug-resistant feature. Since the first case of *C. auris* infection was reported in Japan in 2009, infections with this fungus have been reported in over 40 countries/regions, with mortality rates ranging from 30% to 60% (4). Furthermore, *C. auris* has the potential to cause outbreaks in healthcare settings, particularly in nursing homes for elderly patients, as it efficiently spreads through skin contact. Most importantly, *C. auris* is the first fungal pathogen to demonstrate significant and sometimes untreatable clinical resistance to all known antifungal classes, including azoles, polyenes, and echinocandins (5).

Infections caused by *C. auris* can affect multiple parts of the human body, including the skin, urogenital tract, and respiratory tract. These infections may spread to the bloodstream, leading to extremely high mortality rates (6). The elevated incidence of invasive infections and the significant mortality rate attributed to *C. auris* are linked to their virulence factors, encompassing adhesive properties, biofilm creation, phospholipase, and protease production, as well as their resistance to multiple antifungal agents currently available.

Biofilm formation is one of the major pathogenic traits of *C. auris*. Most of the colonizing and clinically isolated strains exhibit biofilm formation that is equal to or even greater than that of *C. albicans* (7–9). Cell adhesion is a crucial trait enabling pathogens to colonize host cells. By adhering, pathogens acquire the capability to establish microbial communities or biofilms, which constitutes a significant virulence factor for *Candida* species (10). Aggregated phenotype is strongly correlated with clinical *C. auris* isolates, wherein clinical isolates are noted for forming more resilient biofilms (7). During the process of biofilm formation, *C. auris* stimulates the expression of various genes that encode cell wall proteins and adhesins, thereby enhancing the biofilm's ability to adhere to and persist on both biological and non-biological surfaces (11).

Surface colonization factor (Scf1) is an adhesin that is specific to *C. auris*. Scf1 is both necessary and sufficient for the robust attachment of *C. auris* cells to polymer substrates (12). The G1 cyclin gene *Hgc1* not only induces hyphae formation both *in vivo* and *in vitro*, but also contributes to the formation of biofilms (13). The cell wall integrity (CWI) pathway is mediated by the *Mkc1* gene. *Mkc1* is also involved in biofilm formation and is activated through surface contact, which presumably aids invasion into the solid surfaces (14). *Mkc1* can be activated in response to various stresses (15) and plays a role in virulence in the mouse systemic model (16). The enhanced filamentous growth transcriptional factor (Efg1) and the TEA/ATTS transcription factor (Tec1), and Bcr1 can also regulate the formation of biofilms (17). Furthermore, β-1,3-glucan and chitin are fundamental and crucial components of the cell walls of several pathogenic fungi (18). *C. auris* possesses a unique cell wall composition, with an outer mannoprotein layer protecting the inner β−1,3-glucan from immune recognition, thereby enabling immune evasion and drug resistance (19).

*Houttuynia cordata* Thunb. features creeping rhizomes and swollen nodes, primarily distributed in moist and shady areas in East Asian countries such as China, Japan, and Korea (20). The aerial parts of *H. cordata* have long been used in traditional Chinese medicine for the treatment of pneumonia (21), bronchitis (22), dysentery (23), snakebite (24), and antimicrobial and anti-inflammatory (22, 25). The main active ingredient of *H. cordata*, houttuynin (decyl aldehyde), is prone to oxidation or polymerization, leading to the loss of houttuynin. The form used clinically is SH, a compound of houttuynin and sodium bisulfite. SH, which is more stable than houttuynin, retains the main pharmacological activities of houttuynin, including antifungal, antibacterial, and anti-inflammatory activities (26, 27). SNH, also known as sodium dodecylaldehyde bisulfite, is a structural derivative of SH. Both SH and SNH exhibit highly similar structures and anti-infectious biological activities (28). Recent reports have shown that SH and SNH

exhibit an inhibitory effect on *C. albicans*, resulting in reduced overall growth, biofilm formation, metabolic activity, and cell wall remodeling. Previous research found that SNH is effective in inhibiting the adhesion and biofilm formation of *C. albicans*. Importantly, SNH not only suppresses adhesion but also inhibits the formation of a mature biofilm (29). Furthermore, to prevent the formation of biofilms, the transformation of yeast cells to hyphae should be prevented (30) by SNH, which is key to the biofilm maturation of *C. albicans*. However, the effects and mechanisms of SH and SNH against *C. auris* remain unclear.

Here, we perform *in vitro* and *in vivo* assays to investigate the potential of SH and SNH against multidrug-resistant *C. auris* by inhibiting adhesion, biofilm formation, and its reconstruction of the cell wall. The findings of this study may make significant contributions to the treatment options for fungal infectious diseases and may have important theoretical and research value.

## MATERIAL AND METHODS

### *C. auris* strains and cultivation

The *C. auris* clinical strains (C1, C2, C3, C4, 12373, and 12767) were kindly provided by Professor Guanghua Huang from Fudan University. The strains were cultivated on yeast extract peptone dextrose (YPD) agar medium (1% yeast extract, 2% peptone, 2% glucose, and 2% agar) at 37°C, and then activated in YPD liquid medium (1% yeast extract, 2% peptone, and 2% glucose) and incubated at 37°C for 12-14 h until they reached the exponential growth phase. The cultures were centrifuged at 1,467 ×$g$ for 5 min, and the supernatants were discarded to obtain the *C. auris* precipitates. The strains were stored in 50% glycerol at −80°C.

### Molecular identification of *C. auris* strains

The study focused on *C. auris* strains C1, C2, C3, C4, 12373, and 12767. Specific PCR amplification was employed to target the ITS1-ITS4 ribosomal DNA internal transcribed spacer regions of *C. auris* (Table 1). Following agarose gel electrophoresis verification of the amplicons, bidirectional sequencing of the purified DNA fragments was performed, with subsequent sequence assembly via SnapGene software. DNA extraction of *C. auris* was performed using the TIANamp Yeast DNA Kit (DP307-02, Tiangen, Beijing, China). The obtained full-length sequences were submitted to the NCBI database for BLASTn (31).

### Minimum inhibitory concentration determination

According to the CLSI M27-M44S protocol established by the Clinical and Laboratory Standards Institute (CLSI), the broth microdilution method in 96-well plates was used to determine the MIC of six *C. auris* strains against five drugs: SH, SNH, Fluconazole, Amphotericin B, and Caspofungin. And a control was conducted using the *C. parapsilosis* 22019 strain following the CLSI requirement. In this experiment, RPMI-1640 liquid medium was used to adjust the concentration of each fungal suspension to 2 × $10^3$ cells/mL. 100 µL of each drug solution, after serial dilution, was added to the 96-well plate, followed by 100 µL of fungal suspension in each well. A column of blank controls was set up by adding 200 µL of RPMI-1640 liquid medium. The plates were incubated statically at 37°C for 48 h. The MIC of the drugs against *C. auris* strains C1, C2, C3, C4, 12373, and 12767 was observed visually. The MIC represents the minimum drug concentration required for 100% growth inhibition of *C. auris*. The experiment was repeated three times (32).

### Growth curve

The growth curves of *C. auris* strains C1, C2, C3, C4, 12373, and 12767 were determined according to a previous report (33). The research was divided into four groups: a blank

group (no drug added), a 128 µg/mL SH group, a 64 µg/mL SNH group, and a 128 µg/mL Fluconazole (FLU) group. The concentrations of each fungal suspension were adjusted to $2 \times 10^3$ cells/mL using RPMI-1640 liquid medium and placed in 100 mL conical flasks. The flasks were then incubated in a constant temperature shaker at 37°C and 200 rpm. The $OD_{600}$ values of each fungal suspension were measured every 6 h, and the $OD_{600}$ values at different time points were plotted to create the microbial static growth curves for each strain. The experiment was repeated three times.

## Transcriptome sequencing

We adjusted the concentration of *C. auris* strain C1 in the logarithmic growth phase to $2 \times 10^6$ cells/mL using RPMI-1640 medium. Subsequently, *C. auris* was treated with 128 µg/mL SH and 64 µg/mL SNH, respectively. A control group without drug treatment was also included. After incubation at 37°C for 24 h, the fungal suspensions were collected using RNase-free pipette tips, centrifuged at 4°C for 2 min at $9,167 \times g$, and the supernatants were discarded. The precipitates were washed three times with sterile PBS buffer. The samples were then rapidly frozen in liquid nitrogen for 30 min and transferred to a −80°C freezer for storage. Total RNA was extracted from the control group ($n = 3$), SH group ($n = 3$), and SNH group ($n = 3$) for transcriptome sequencing. The concentration, purity, and integrity of the extracted RNA were assessed using a Nanodrop system and agarose gel electrophoresis. mRNA was isolated from total RNA through A-T base pairing with Oligo(dT)-coated magnetic beads targeting the poly-A tail at the 3′ end of mRNA. A fragmentation buffer was added to randomly fragment the mRNA, which averaged several kb in length, and magnetic beads were used to select small fragments of approximately 300 bp. With the addition of random hexamers, reverse transcription was initiated to synthesize the first-strand cDNA using mRNA as the template. Second-strand synthesis was then performed to form a stable double-stranded structure. Since the double-stranded cDNA has sticky ends, End Repair Mix was added to convert them into blunt ends. An "A" base was added to the 3′ ends for ligation to a Y-shaped adapter. Sequencing was performed on the Illumina platform: after 15 cycles of PCR amplification to enrich the library, the target bands were recovered and mixed in proportion for sequencing. Bridge PCR amplification was performed on the cBot to generate clusters, followed by sequencing (PE library, reads $2 \times 150$ bp). The analysis was performed on BioCloud Analytics of Qingke Biology. These sequence data have been submitted to the NCBI SRA under the BioProject accession number PRJNA1208975 (34).

## Adhesion assay

The adhesion activities of *C. auris* strains C1, C2, C3, C4, 12373, and 12767 were determined. Based on the previous report (35), the slides, each possessing a surface area of 1 cm², were immersed in a 75% ethanol solution for a duration of 24 h. Subsequently, they were thoroughly rinsed with sterile PBS. A suspension containing *C. auris* at a concentration of $2 \times 10^6$ cells per milliliter was then introduced into a six-well plate, accompanied by 128 µg/mL SH, 64 µg/mL SNH, and 128 µg/mL FLU. The plates were incubated at a temperature of 37°C for a period of 3 h. Afterward, the slides were stained in the dark for 30 min using calcofluor white (CFW). Fluorescent expression was observed using a Stellaris 5 Cryo (Leica, Wetzlar, Germany).

## Aggregation assay

The aggregation activities of *C. auris* strains C1, C2, C3, C4, 12373, and 12767 were determined. Based on the previous report (36), the concentration of *C. auris* during its logarithmic growth phase was adjusted to $2 \times 10^8$ cells/mL. Subsequently, 128 µg/mL SH, 64 µg/mL SNH, and 128 µg/mL FLU were added to the *C. auris* suspension. A control group that received no treatment was also included. The fungal suspensions underwent incubation at 37°C for 120 min. Subsequently, the liquid was examined with a confocal microscope of Stellaris 5 Cryo (Leica, Wetzlar, Germany).

**TABLE 1** Primer sequences used for PCR

| Primer name | Sequence (5′ to 3′) |
| --- | --- |
| ITS1 | TCCGTAGGTGAACCTGCGG |
| ITS4 | TCCTCCGCTTATTGATATGC |

## XTT assay of biofilm

The 2,3-Bis(2-methoxy-4-nitro-5-sulfophenyl)–2H-tetrazolium-5-carboxanilide (XTT) assay (37) was employed to assess the ability of the drug to inhibit biofilm formation of *C. auris* strains C1, C2, C3, C4, 12373, and 12767. The concentrations of each fungal suspension were adjusted to $2 \times 10^3$ cells/mL using RPMI-1640 medium. 100 µL of each fungal suspension was added to a 96-well plate. After static incubation at 37 °C for 90 min, the supernatants were carefully aspirated, and 100 µL of PBS buffer was added to each well. The plate was gently oscillated, and the washing process was repeated three times. Subsequently, 100 µL of RPMI-1640 liquid medium was added to each well, gently oscillated, and then incubated statically at 37°C for an additional 24 h. The supernatants were carefully aspirated again, and 100 µL of PBS buffer was added to each well. The plate was gently oscillated, and the washing process was repeated three times. Then, 150 µL of XTT was added to each well in the dark. After incubation for 3 h, the absorbance of each well was measured at 490 nm. The experiment was repeated three times.

## Crystal violet assay of biofilm

The biofilm formation activities of *C. auris* strains C1, C2, C3, C4, 12373, and 12767 were determined by crystal violet staining. The concentrations of each fungal suspension were adjusted to $2 \times 10^6$ cells/mL using RPMI-1640 medium. 128 µg/mL SH, 64 µg/mL SNH, and 128 µg/mL FLU were added separately. A control group without drug administration was also set up, and the drug solutions were co-incubated with the fungal suspensions for 24 h. The fungal cells were washed three times with sterile PBS to remove non-adherent cells. The cells were then incubated with crystal violet staining solution (Yeasen Biotechnology Co., Ltd., Shanghai, China) for 45 min, washed twice with sterile PBS, and subsequently incubated with 95% (vol/vol) ethanol for 45 min. Finally, the absorbance of each well was measured at 560 nm using a microplate reader (38).

## RT-qPCR

Total cDNA extracted from *C. auris* strains C1 and C2 was used to synthesize cDNA following the instructions of the manufacturer for the reverse transcription kit (Yeasen Biotechnology Co., Ltd., Shanghai, China). Table 2 depicts the primer sequences. The

**TABLE 2** Primers sequences used for RT-qPCR

| Primer name | Sequence (5′ to 3′) |
| --- | --- |
| *Bcr1* F | CCGCCACCGCCGTAAACAC |
| *Bcr1* R | CCGTCTGCTTGTGCTGTCTGAG |
| *Tec1* F | AAATGCCCACGCTGTCTCACTTC |
| *Tec1* R | GCTGCTCGTCTAAGTTGGAGTCTG |
| *Efg1* F | CAGCACCAGCAGCAGCAGTAC |
| *Efg1* R | GCAGCAGAAGAGTTGGAGTAACCG |
| *Mkc1* F | ATGCTAACCCTTTGGCTCTTGACC |
| *Mkc1* R | AATGCTTCGTCCACGGTGATTCTC |
| *Hgc1* F | AACAACAACAACAACAACAACAACATC |
| *Hgc1* R | TGATGATGATCTTGAATTGGCATAACC |
| *Scf1* F | CCAAAGGGTGAACAGCCAGAAGG |
| *Scf1* R | ACAGAAGGAGCAGGAGCAGGTC |
| *Actin* F | GGCTCATCTTGGCTTCCTT |
| *Actin* R | GGACCGGACTCGTCGTATTC |

RT-qPCR amplification was conducted using a Light Cycler 96 for fluorescence quantification. Gene expression levels were analyzed using the relative quantification method based on the cycle threshold values ($2^{-\Delta\Delta Ct}$)

## Confocal laser scan microscope assay

The study focused on *C. auris* strains C1, C2, C3, C4, 12373, and 12767. The concentration of *C. auris* in the logarithmic growth phase was adjusted to $2 \times 10^6$ cells/mL. Subsequently, the *C. auris* strains were treated with 128 µg/mL of SH, 64 µg/mL of SNH, and 128 µg/mL of FLU, respectively. A control group without any drug treatment was also included. After incubation at 37°C for 24 h, non-adherent fungal cells were removed by rinsing with sterile PBS. Subsequently, Calcofluor White (CFW, 10 µM, Sigma-Aldrich, Shanghai, China) staining solution was added, and the samples were stained in the dark for 5 min. Fluorescent expression was observed using a Stellaris 5 Cryo CLSM (Leica, Wetzlar, Germany).

## Detection of cell wall components

The cell wall components of *C. auris* strain C1 were determined based on a previous report (39). Fungal cells ($2 \times 10^6$ cells/mL) were treated with 128 µg/mL SH, 64 µg/mL SNH, and 128 µg/mL FLU, respectively, at 37°C for 24 h. Subsequently, the fungal cells were collected, washed with sterile PBS, and treated with a suspension of the fungi in calcofluor white (CFW, 10 µM, Sigma-Aldrich, Shanghai, China) for dark staining to assess chitin content. As previously described, the samples were blocked with 3% bovine serum albumin for 1 hour, followed by centrifugation at 825 g for 5 min and washing three times with PBS. Subsequently, the cells were incubated with a monoclonal anti-β-glucan antibody (1:300, 4002, Bioscience Supplies, Australia) diluted in PBS at 4°C for 1.5 h. After primary antibody treatment, the fungal cells were washed three times with PBS and incubated with diluted Cy3-labeled goat anti-mouse IgG (1:100, A22210, Abbkine, Shanghai, China) at 4°C for 20 min. The cells were stained in the dark with 5 µM of the plasma membrane-specific fluorescent probe PM-1 (1449483-78-6, MedChemExpress, USA) for 20 minutes. Finally, the cells were washed three times with PBS, and the average fluorescence intensity of 10,000 fungal cells was detected by flow cytometry. Fluorescent expression was observed using a Stellaris 5 Cryo (Leica, Wetzlar, Germany).

## *G. mellonella* larvae infection model

In this experiment, 90 agile *G. mellonella* larvae, each with a body length of 2.0–2.5 cm, were selected. The *C. auris* strains C1 and C2 strains, with higher growth rates, were cultivated overnight in YPD liquid medium at 37°C with shaking at 200 rpm for the larvae infection. After being washed twice with PBS, the fungal suspension's concentration was adjusted to $8 \times 10^6$ cells/mL. A 10 µL aliquot of the fungal suspension was injected into the left hind leg using a sharp microinjection needle. Two hours later, sterile PBS, 128 µg/mL SH, 64 µg/mL SNH, and 128 µg/mL FLU were injected into the right hind leg using the same method. The larvae were then incubated at 37°C. After 48 h, three larvae from each group were randomly selected, homogenized, and plated onto YPD solid plates. The plates were incubated upside down at 37°C for 48 h to observe the fungal load results. In addition, the larvae's mortality was observed every 24 h to plot a survival rate curve, and survival analysis was conducted using the Log-rank Test.

## Mouse model of systemic *C. auris* infection

All procedures involving animals were approved by the Animal Ethics Committee, Institute of Anhui University of Chinese Medicine. Maintenance and treatment of all animals complied with the principles of the Institutional Animal Ethics Committee of the Chinese Center for Disease Control and Prevention and conformed with the Chinese National Guidelines on the Care and Use of Laboratory Animals. Based on the previous report (40), a total of 135 female Balb/C mice (aged 6–7 weeks, weighing 20 ± 2 g) were

adaptively fed with standard granular feed and tap water for 7 days in a room with a temperature of 23°C ± 2°C, a relative humidity of 40%–70%, and an alternating 12 h light and 12 hour dark cycle. In the second week, all mice were randomly and equally divided into nine groups: PBS group, model group, FLU group, SH group (high, medium, and low doses), and SNH group (high, medium, and low doses), with 15 mice in each group. On days −4, −1, and +2 relative to infection, the mice were injected intraperitoneally with 150 mg/kg and 100 mg/kg of cyclophosphamide for immunosuppression. According to the different groups, the *C. auris* strain C1 fungal suspension adjusted to $3 \times 10^7$ cells/mL with sterile PBS and an equal volume of sterile PBS were injected into the tail veins of the mice at a dose of 100 µL per mouse. Two h post-modeling, daily doses of 150, 100, and 50 mg/kg of SH and SNH, respectively, along with 2 mg/kg of FLU, were administered to the mice. Starting from the first day after successful modeling, the general condition of the mice was observed daily, and the survival time was recorded. After 7 days of observation, the survival percentage was calculated, a survival curve was plotted, and survival analysis was performed using the Log-rank Test.

## *C. auris* load in mouse liver/kidney tissue

At 72 h post-modeling, three mice from each group were euthanized by cervical dislocation, and their left kidneys were removed under sterile conditions and fixed in 4% paraformaldehyde for at least 24 h. The liver tissue and right kidney were rinsed with sterile saline, weighed, placed in a homogenizer, and mixed with 1 mL of sterile PBS for thorough grinding. The homogenate was then serially diluted in sterile PBS at a 10-fold ratio across four concentration gradients and plated onto YPD medium containing 100 U/mL penicillin and 0.1 mg/mL streptomycin. After incubation at 30°C for 48 h, the colonies were observed and counted, and the average colony count was calculated using the formula: $\log_{10}$ CFU/g = $\log_{10}$ ([average colony count from three plates × dilution factor]/organ weight).

## Pathological examination

The inflammatory response in the target organs of mice with systemic *C. auris* infection was assessed using an optical microscope. At 72 h post-infection (day 3), three surviving mice from each group were euthanized, and tissues were excised and fixed in 10% buffered formalin. The fixed tissues underwent tissue processing, embedding in paraffin, sectioning, and staining with hematoxylin and eosin (H&E) and periodic acid-Schiff (PAS).

## Immunohistochemistry

Liver and kidney tissue sections were deparaffinized, hydrated, and had endogenous enzymes blocked. After occlusion with goat serum, the sections were incubated overnight at 4°C with primary antibodies (IL-6, TNF-α, and IL-10). Subsequently, the sections were washed with PBS and incubated with secondary antibodies at 37°C for 40 minutes. Following additional PBS washes, horseradish peroxidase-labeled streptavidin protein was added and incubated at 37°C for 40 minutes. After thorough PBS washes, the sections were visualized using a digital panoramic scanner (WS-10, WISLEAP, China) after DAB development. Finally, quantitative immunohistochemical analysis of inflammatory factors in mouse liver and kidney tissues was conducted based on Image J software (41).

## Statistical analysis

GraphPad Prism 9.0 was utilized for conducting the statistical analysis. The information is provided from a minimum of three distinct trials and represented as the mean ± standard deviation (SD) ($n \geq 3$). To compare two groups of data, the student's t-test was employed. In addition, for comparing multiple groups of data, the two-way ANOVA test was applied. Significant differences were considered for *P*-values below 0.05.

## RESULTS

### SH and SNH exhibit significant inhibitory effects on the growth of *C. auris*

In this study, the internal transcribed spacer (ITS) sequences of six fluconazole-resistant *C. auris* strains, namely C1, C2, C3, C4, 12373, and 12767, were obtained through PCR amplification, and a phylogenetic tree was constructed based on the ITS sequences. The molecular identification results (data not shown) showed that six clinically isolated strains were confirmed as *C. auris*. In combination with drug-resistant phenotypic identification and collaborative communication by Professor Huang Guanghua's team at Fudan University indicated that the aforementioned strains all belonged to the South Asia Clade (Clade I) of *C. auris*. First, we evaluated the growth inhibitory effects of SH and SNH on *C. auris* by determining their MIC values against six fluconazole-resistant *C. auris* strains. The results, as shown in Table 3, indicate that the MIC values of FLU against *C. auris* ranged from 128 to 1024 µg/mL, while the MIC values of SH and SNH against *C. auris* were both within the range of 32–128 µg/mL. This suggests that SH and SNH exhibit stronger antimicrobial effects on the drug-resistant *C. auris* strains. Subsequently, we investigated the growth inhibitory effects of SH and SNH on *C. auris* at different time intervals. These results (Fig. 1A through D) demonstrate that cells in groups C1 and C2 exhibited normal growth curves, with an active exponential phase from 18 to 30 h and a plateau phase from 30 to 72 h. We found that FLU failed to inhibit the growth of C1 and C2 after 66 h, whereas SH and SNH were still able to inhibit their growth after 72 h. Similarly, cells in groups C3 and C4 showed normal growth curves, with an active exponential phase from 30 to 48 h and a plateau phase from 48 to 72 h. SH failed to inhibit the growth of C3 and C4 after 54 and 60 h, respectively, whereas SNH continued to inhibit their growth after 72 h. In addition, we examined the growth inhibition effects of SH and SNH on the 12767 and 12373 strains, with the results presented in the figures (Fig. 1E and F). We observed that the cells in the 12767 and 12373 groups exhibited normal growth curves, with active exponential phases occurring from 42 to 60 h and 60 to 78 h, respectively, and plateau phases from 66 to 96 h and 84 to 96 h, respectively. However, for the 12767 strains, FLU failed to inhibit its growth after 78 h, whereas SH and SNH were still able to suppress its growth at 96 h. For the 12373 strains, both SH and SNH were unable to inhibit their growth after 96 h, while FLU continued to exert an inhibitory effect at 96 h. Therefore, compared with the control group, significant growth delays were observed in the SH and SNH groups. These results indicate that SH and SNH have potent fungistatic effects on drug-resistant *C. auris* strains, with SNH showing a stronger inhibitory effect.

### Effect of SH and SNH on the transcriptomics profile of *C. auris*

To further understand the potential molecular mechanisms of SH and SNH against *C. auris*, we conducted transcriptome sequencing analysis on *C. auris* samples from the control group and the SH and SNH treatment groups using the Illumina high-throughput sequencing platform. Principal component analysis (Fig. 2C) revealed significant differences between these groups, indicating that pharmacological interventions can

**TABLE 3** MIC of SH, SNH, FLU, AmB, and CAS against *C. auris*[a,b]

| Strain | SH | SNH | FLU | AmB | CAS |
|---|---|---|---|---|---|
| C1 | 128 µg/mL | 64 µg/mL | 1024 µg/mL | 2 µg/mL | 4 µg/mL |
| C2 | 128 µg/mL | 64 µg/mL | 1024 µg/mL | 1 µg/mL | 4 µg/mL |
| C3 | 128 µg/mL | 64 µg/mL | 1024 µg/mL | 1 µg/mL | 4 µg/mL |
| C4 | 128 µg/mL | 64 µg/mL | 1024 µg/mL | 1 µg/mL | 4 µg/mL |
| 12373 | 64 µg/mL | 32 µg/mL | 128 µg/mL | 0.25 µg/mL | 0.5 µg/mL |
| 12767 | 64 µg/mL | 32 µg/mL | 128 µg/mL | 2 µg/mL | 0.5 µg/mL |
| *C. parapsilosis* 2201 | N/A | N/A | N/A | 0.5 µg/mL | 0.5 µg/mL |

[a]FLU means Fluconazole, AmB means Amphotericin B, and CAS means Caspofungin.
[b]N/A, Not applicable.

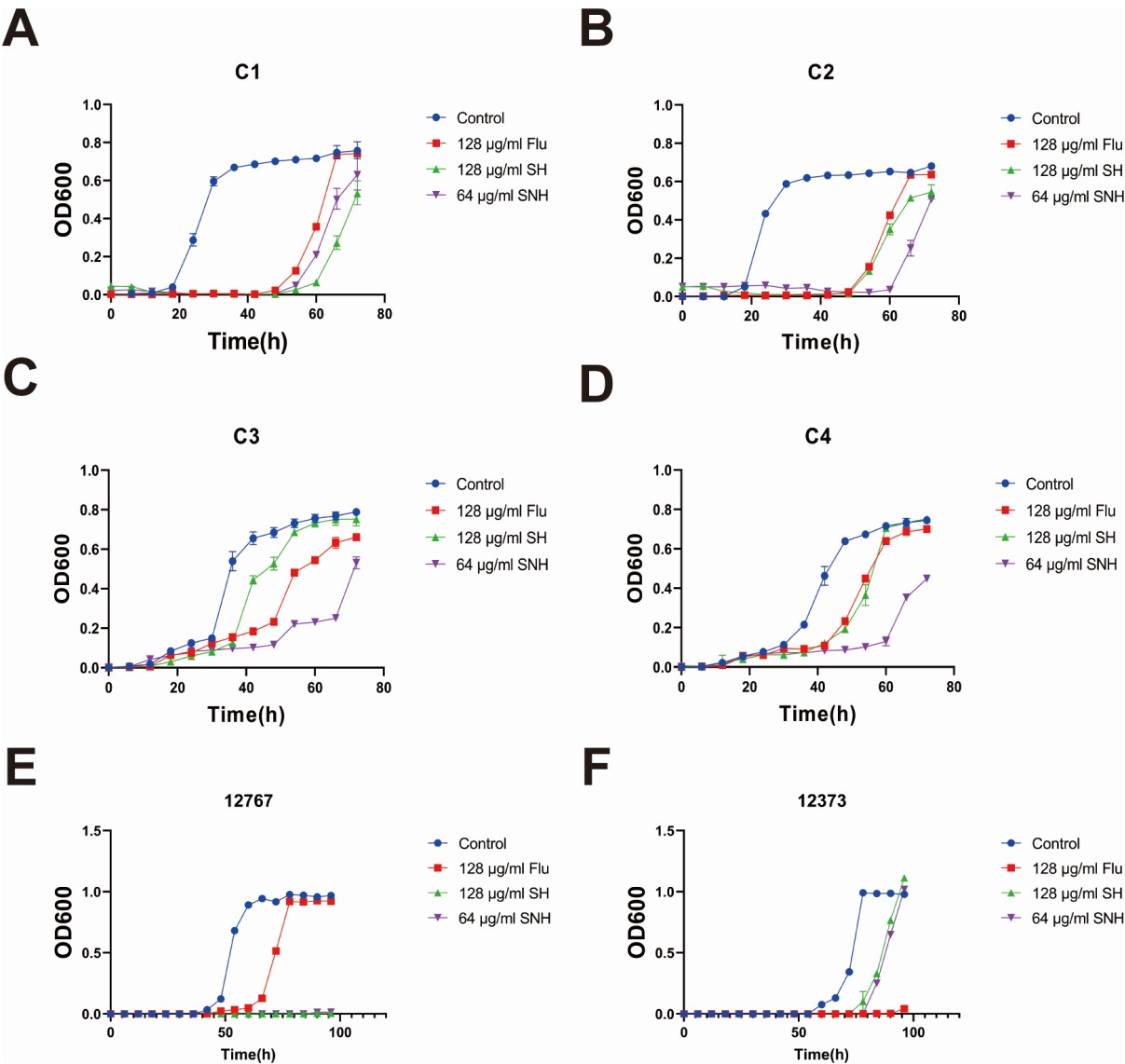

**FIG 1** SH and SNH exhibit significant inhibitory effects on the growth of *C. auris*. (A through F) Growth inhibition curves of SH and SNH against six drug-resistant *C. auris* strains. Note: FLU means fluconazole.

affect transcriptional patterns. The focus of this study is on the genes that exhibit differential expression in the SH and SNH treatment groups compared to the control group, while 532 differentially expressed genes were identified in the SNH treatment group, including 246 upregulated genes and 286 downregulated genes (Fig. 2A and B). Finally, we utilized GO barplot analysis to further explore the impact of SH and SNH on the functions of *C. auris* in terms of molecular function (MF), cellular component (CC), and biological process (BP). We found (Fig. 2D) that the pathways enriched by the majority of genes included transmembrane transporter activity and peroxisomal and fatty acid beta-oxidation. Through the comparison of transcriptome sequencing results, we also found that the expression of some genes related to the adhesive growth and aggregation of *C. auris* was downregulated, including *Scf1*, which encodes the *Candida* adhesin gene, as well as *Hgc1*, which is involved in *Candida* morphological transition and pathogenicity. This suggests that SH and SNH interventions significantly altered the gene expression profile of *C. auris*, particularly those genes related to aggregation and biofilm formation abilities, which are associated with pathogenicity.

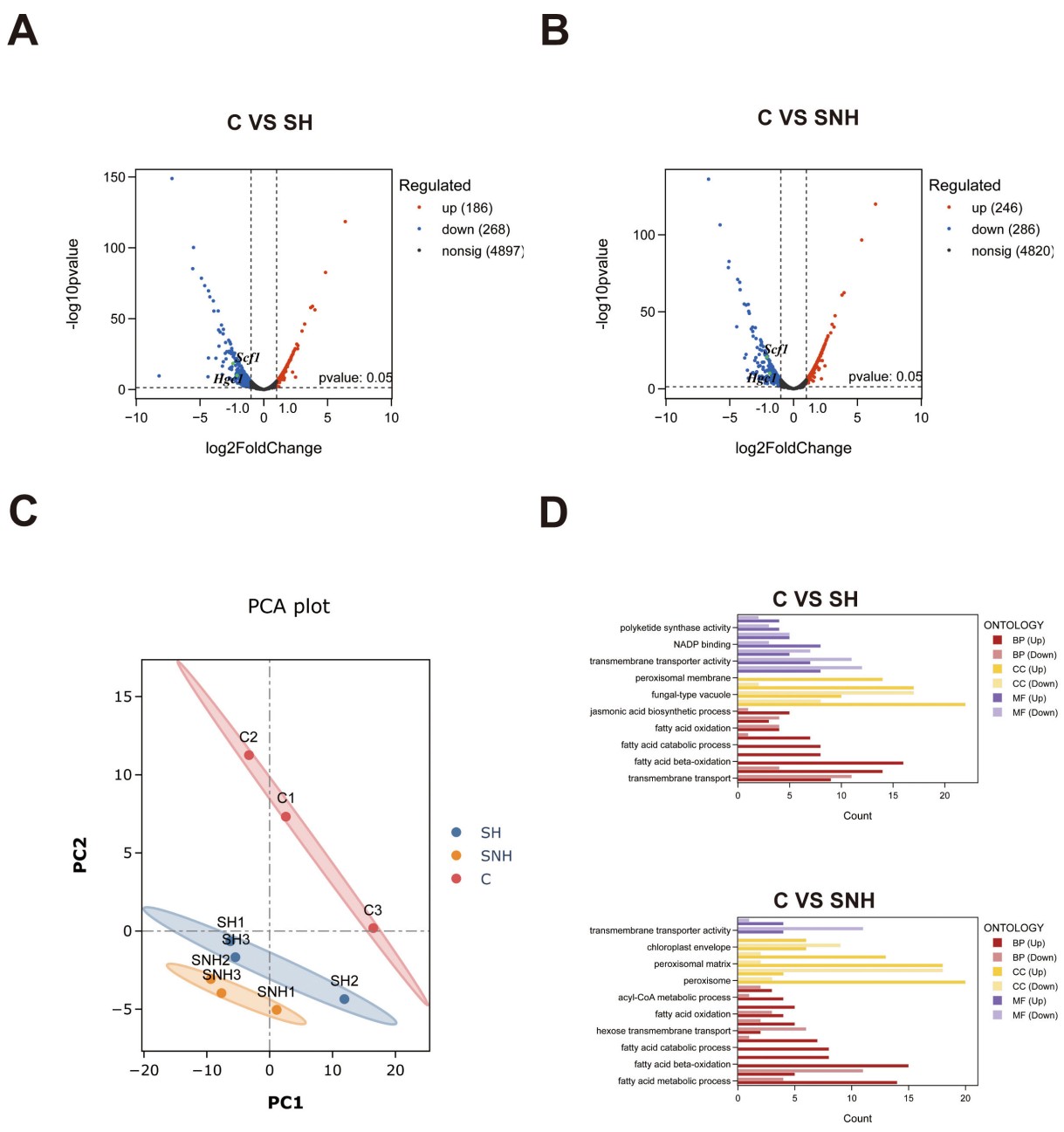

**FIG 2** Effect of SH and SNH on the transcriptomics profile of *C. auris*. (A) Volcano plots comparing the SH group and the control group. (B) Volcano plots comparing the SNH group and the control group. (C) Principal components analysis (PCA) plot depicting the samples from the control, SH, and SNH groups. (D) GO bar chart representation of control, SH, and SNH samples. Note: C means control group, SH means SH group, and SNH means SNH group.

## SH and SNH inhibit the adhesion and aggregation of *C. auris*

The ability of *C. auris* to form biofilms on various surfaces enhances its robustness and starts with adhesion and aggregation (42). The results (Fig. 3A and B) of fluorescence microscopic observation and quantitative analysis indicate that after a 3-hour incubation, only a small number of *C. auris* cells adhered to the cover glasses. By contrast, compared to the control group and the FLU group, the number of *C. auris* cells adhering to the cover glasses in the SH group and the SNH group was significantly reduced. These findings suggest that SH and SNH can significantly inhibit the early adhesion of *C. auris*. *C. auris* can also aggregate under certain conditions, affecting its biofilm formation, drug sensitivity, and pathogenicity (43). Microscopic observation results showed that (Fig. 3C),

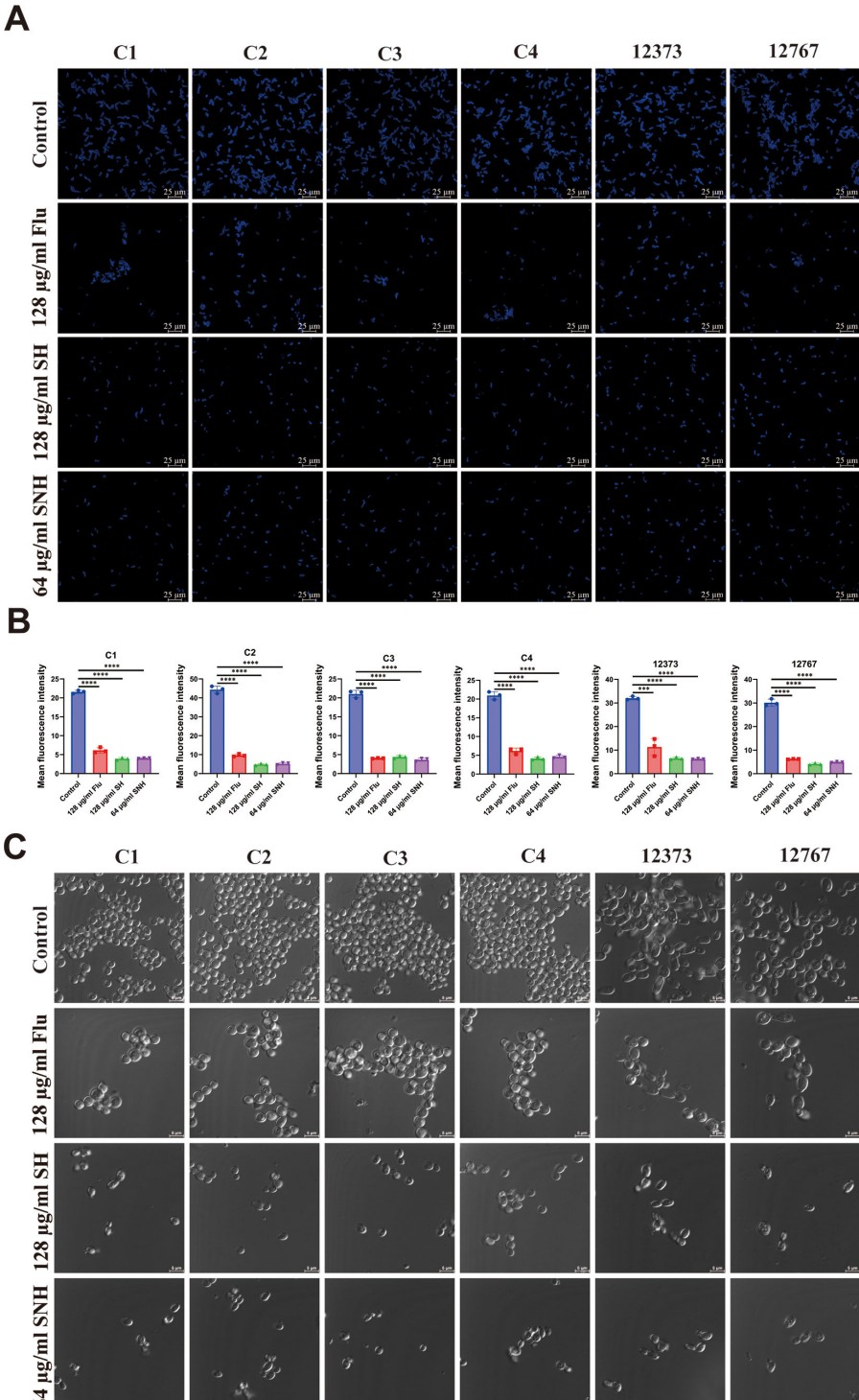

**FIG 3** SH and SNH inhibit the adhesion and aggregation of *C. auris*. (A) Images represent the state of *C. auris* adhesion at 3 h. Scale bars 25 µm. (B) Quantification of *C. auris* number by Image J. (C) Images represent the aggregation of *C. auris* following treatment with SH and SNH. Scale bars 5 µm.

after 2 h of incubation, only a small amount of *C. auris* aggregated at the bottom of the petri dishes in the SH and SNH groups, compared to significant aggregation in the control and FLU groups. These results indicate that SH and SNH can significantly inhibit

the aggregation of *C. auris*. Based on these findings, we conclude that SH and SNH can effectively inhibit the early growth of *C. auris*.

## SH and SNH inhibit biofilm formation by *C. auris*

Biofilm, a structured microbial community anchored by a protective matrix, can form from single or mixed cultures of hyphal cells and yeast forms (44). As a therapeutic refuge, it initiates or prolongs infections, allowing cells to infiltrate tissues and create new infection foci (45). As mentioned above, transcriptome results indicate that SH and SNH can significantly impact the biofilm-forming capacity of *C. auris*. Therefore, we evaluated the inhibitory effects of SH and SNH on *C. auris* biofilms by assessing their biofilm-forming ability using the XTT assay. We detected the biofilm formation of six *C. auris* strains using the XTT staining method. The results, as shown in Fig. 4A, indicate that the OD values of *C. auris* measured at 490 nm suggest that SH and SNH can hinder the biofilm formation of *C. auris* to a certain extent. Subsequently, we used crystal violet staining to detect the biofilms and assessed the staining level of *C. auris* on the bottom of a 96-well plate. The results are shown in Fig. 4B, with OD values measured at 560 nm using a microplate reader. Purple residues on the bottom of the 96-well plate indicate the presence of *C. auris* biofilms stained with crystal violet. The control group showed intense crystal violet staining on the bottom of the plate, indicating vigorous growth of *C. auris* biofilms. By contrast, the wells treated with SH and SNH showed reduced crystal violet staining. The OD values at 560 nm confirm that SH and SNH inhibited the growth of *C. auris* biofilms. The inhibitory effects of SH and SNH on *C. auris* biofilm formation were further investigated using a confocal laser scan microscope (CLSM). As shown in Fig. 4C, the biofilm structure in the control group remained intact, with fungal cells in good condition, vigorous growth, and thicker biofilm formation. However, the biofilm thickness and cell number were reduced in the SH and SNH groups, indicating the absence of complete biofilm formation. Based on these findings, we conclude that SH and SNH can effectively inhibit the biofilm formation of *C. auris*.

## SH and SNH mildly disrupt the structure of the *C. auris* cell wall

The structure and complexity of *C. auris* cell wall components facilitate interactions between the host and the fungus, playing a pivotal role in virulence while enhancing the fungus's resistance to the innate immune system (19). *C. auris* can alter its cell wall composition and structure in response to environmental changes. Figure 5A shows that chitin is uniformly distributed on the cell surface before treatment with SH and SNH. However, after treatment with SH and SNH, a higher concentration of chitin is observed in the outer layer of *C. auris* cells. Analysis of fungal cell fluorescence intensity by flow cytometry (Fig. 5B) reveals that, although not as significant as the effect of FLU, the content of exposed chitin on the surface of *C. auris* increases after treatment with SH and SNH. In addition, the results of Fig. 5C show that after treatment with SH and SNH, the exposure of glucan in the fungal cell wall increases, as indicated by staining with a fluorescent dye and analysis of fluorescence by flow cytometry (Fig. 5D). Through combined PM-1 and CFW staining analysis, we found that the chitin deposition level in the cell wall at the budding site of the untreated control group strain C1 was relatively low. By contrast, after treatment with FLU, SH, and SNH, the newly formed bud regions of *C. auris* exhibited a significant enhancement in chitin deposition (Fig. 5E). These results are similar to those obtained from the chitin assay. These findings suggest that SH and SNH might mildly alter the structure of the *C. auris* cell wall.

## SH and SNH inhibit the expression of virulence factor genes in *C. auris*

In *C. auris*, the ability of cell adhesion and aggregation is closely related to its virulence, and hyphal growth is a key factor in adhesion and aggregation. By comparing the results of transcriptome sequencing, we found that the expression of some genes related to adhesion, aggregation, and hyphal growth in *C. auris* was downregulated. These genes

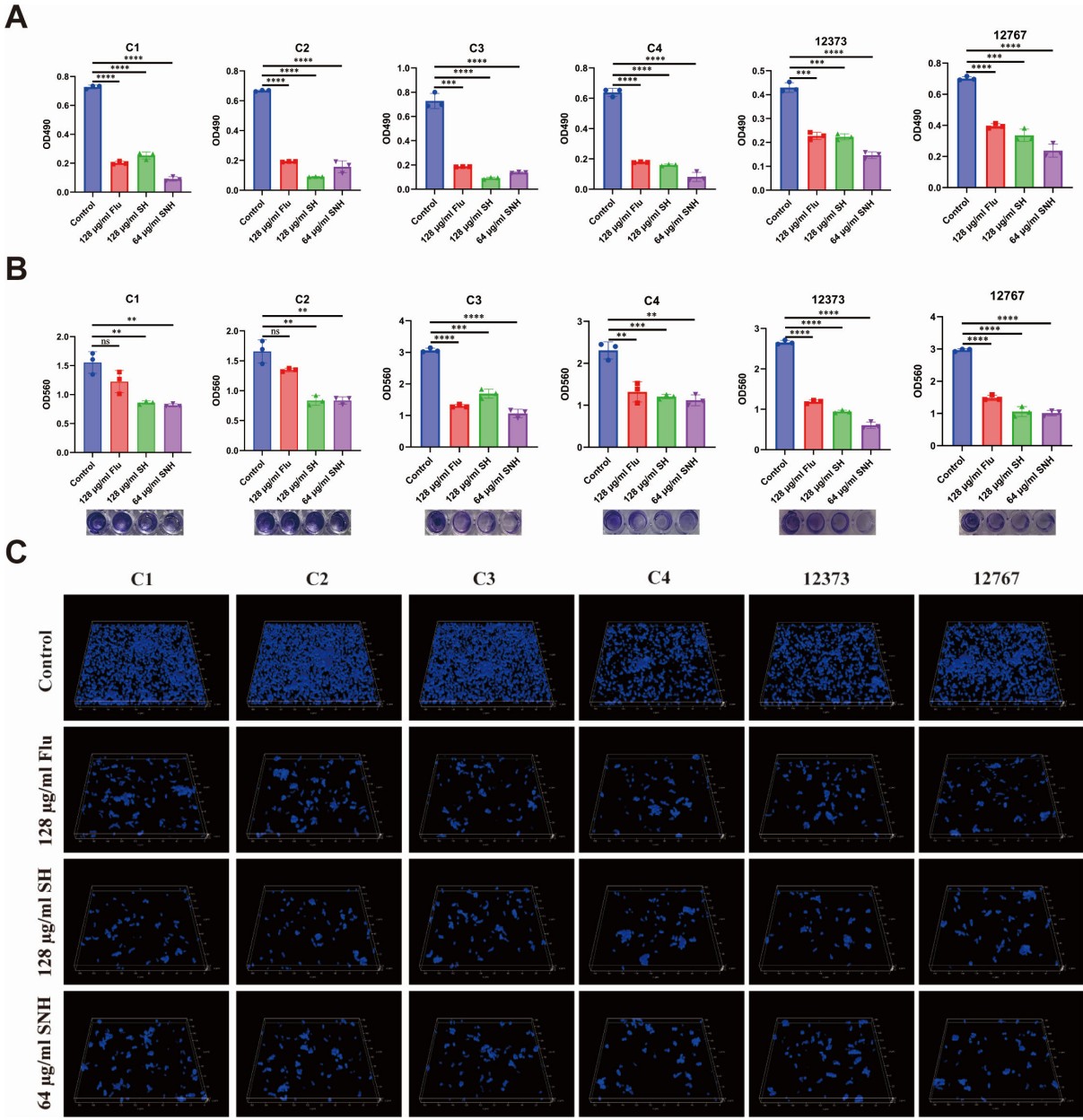

**FIG 4** SH and SNH inhibit biofilm formation of *C. auris*. (A) Images represent the effect of SH and SNH on the metabolic viability of *C. auris*. (B) Images represent the effect of SH and SNH on the biomass of *C. auris* biofilms. The biomass of biofilms formed by *C. auris* was measured in 96-well plates using crystal violet staining. (C) Images represent the morphological structure of *C. auris* biofilms under CLSM observation. 3D views of the CLSM images were generated using Stellaris 5 Cryo software and 3D reconstruction techniques.

include the hypha-regulating factor *Bcr1*, the central transcriptional regulator *Efg1*, the mating inhibition factor *Tec1*, the mitogen-activated protein kinase *Mkc1*, the specific adhesin *Scf1*, and the G1 cyclin gene *Hgc1*. Subsequently, we used real-time PCR to detect changes in the transcription levels of six key virulence-related genes in the drug-resistant strains C1 and C2 of *C. auris* at the mRNA level. The results, as shown in Fig. 6A and B, indicate that compared to the Control group, the expression levels of the six virulence genes in C1 and C2 treated with SH and SNH were downregulated in terms of mRNA expression. Thus, these results suggest that SH and SNH can effectively inhibit the expression of virulence factor-related genes in *C. auris*.

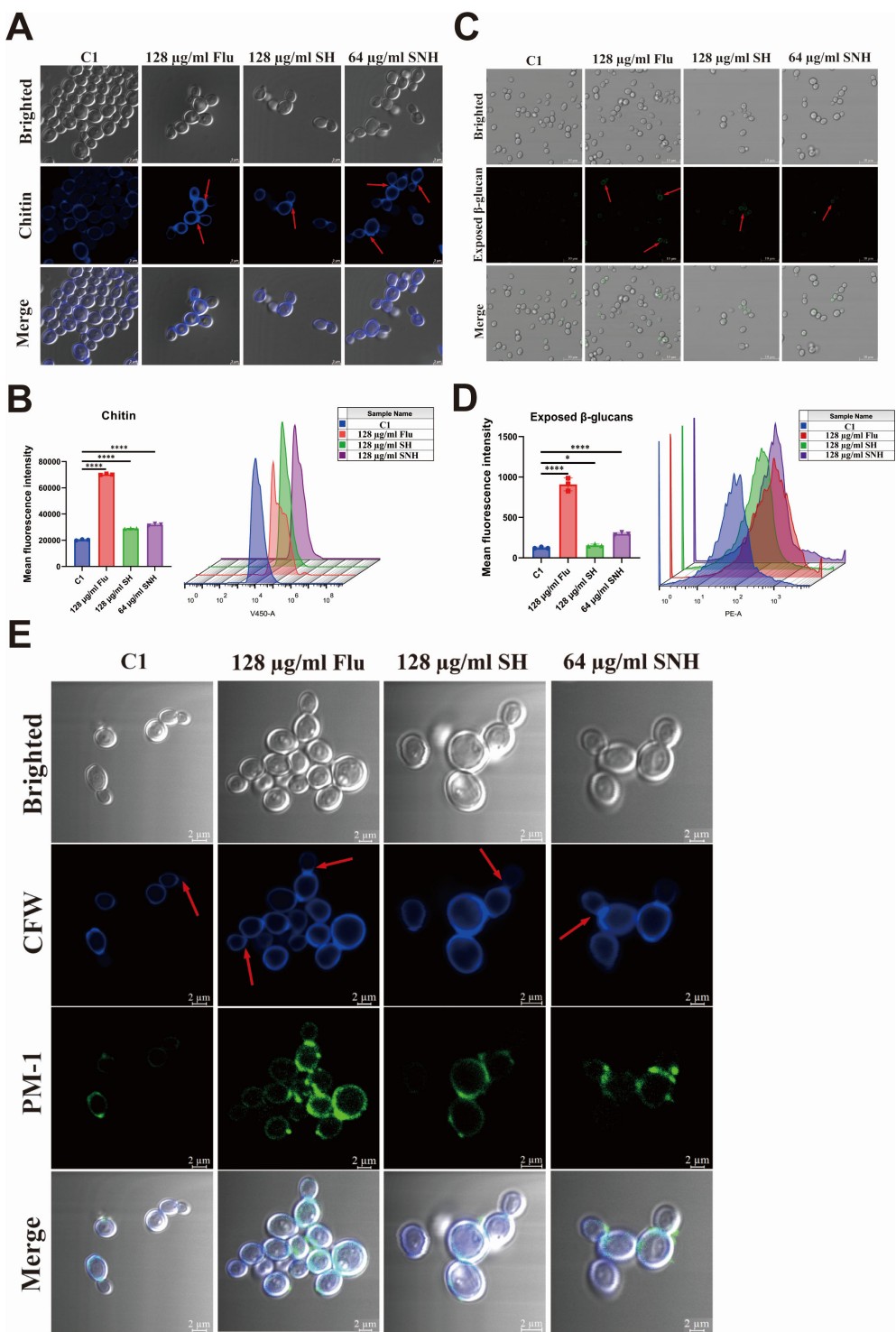

**FIG 5** SH and SNH affect the cell wall structure of *C. auris*. (A) Representative microscopic images of chitin. Scale bar: 5 µm. (B) Average fluorescent intensity (left) and representative flow cytometry profiles (right) of the total amount of chitin assessed by CFW. (C) Representative microscopic images of β-glucan exposure. Cells were cultured overnight in YPD medium and stained with anti- (1, 3)-glucan antibody and Cy3-conjugated secondary antibody. Scale bar: 10 µm. (D) Average fluorescent intensity (left) and representative flow cytometry profiles (right) of CW β-glucan exposure. Cells were incubated with primary anti-β-glucan antibody and PE-conjugated secondary antibody. (E) Representative microscopic images of plasma membrane and cell wall by PM-1 and CFW staining, respectively. Scale bar: 2 µm.

## SH and SNH inhibited the infection of *C. auris* in *G. mellonella* larvae

Based on the above results, we found that SH and SNH treatments weaken the biofilm formation of *C. auris*, thereby effectively reducing its virulence. Therefore, we first used invertebrate *G. mellonella* larvae, which have a natural immune system close to mammals, to assess *C. auris* infections and investigate whether SH and SNH can reduce the virulence of *C. auris* strains C1 and C2 *in vivo*. First, we conducted a 5-day survival observation on *G. mellonella* larvae infected with C1 and C2. We injected $2 \times 10^6$ cells of *C. auris* strains C1 and C2, along with an equal volume of PBS, FLU, SH, and SNH, into the larvae and observed their survival status daily. The results, shown in Fig. 7A, indicate that on the fifth day, all 10 *G. mellonella* larvae in the PBS control group survived. The survival rates of larvae in the C1 and C2 model groups were 30% and 40%, respectively, while the survival rates of larvae in the SH and SNH groups were 80%, 70%, and 90%, 80%, respectively. In addition, the fungal load results shown (Fig. 7B) demonstrate that compared to the model group, the fungal load in the larvae was significantly reduced and exhibited a concentration-dependent pattern. These findings suggest that SH and SNH can effectively treat *C. auris* infections in *G. mellonella* larvae *in vivo*.

## SH and SNH inhibited the systemic infection of *C. auris* in mice

To date, several animal models for systemic *C. auris* infections have been published (46), including mammalian models (47–50). To emulate human diseases typically caused by the penetration of physical barriers of *C. auris*, we further utilized systemic infection of *C. auris* in mice to determine the *in vivo* anti-infection effects of SH and SNH against *C. auris*. The results, shown in Fig. 8A and B, indicate that all mice in the model group died on day 4, and all mice in the FLU group died within 6 days. Notably, mice in the groups receiving high and median dosages of SH and SNH had survival rates of 33.3%, 22.3% for SH, 44.4%, 22.3% for SNH, on day 7. Simultaneously, the fungal load in the liver and kidney of the mice was examined, with the results shown in Fig. 8C and D. Compared to the model group, the fungal load in the liver and kidneys of mice in the groups receiving high and median dosages of SH and SNH was significantly reduced compared to the model group. PAS staining of the liver, kidney, and spleen of the mice (Fig. 8E through G) revealed a large number of *C. auris* aggregations in the liver, kidney, and spleen of the model group, while only a few *C. auris* aggregations were observed in the liver, kidney, and spleen of mice in the SH and SNH treatment groups. Finally, H&E staining of the liver, kidney, and spleen of the mice (Fig. 8H through J) showed extensive patchy necrosis in the kidney and spleen, and a large number of inflammatory cell infiltrations in the liver tissue of the model group. By contrast, the kidney and spleen tissues of mice in the SH and SNH treatment groups were not extensively damaged, and only a small number of inflammatory cell infiltrations were observed in the liver. In summary, the results from the mouse infection model suggest that SH and SNH can effectively treat systemic *C. auris* infections in mice.

## SH and SNH reduced the inflammatory response to systemic infection of *C. auris* in mice

The most common inflammatory factors include interleukins (IL), tumor necrosis factors (TNF), and interferons (IFN), among others (51). The levels of inflammatory factors provide the most direct indication of the severity of inflammation in mice infected with an inflammatory condition. Therefore, we selected two representative pro-inflammatory factors (interleukin-6 and tumor necrosis factor-alpha) and one anti-inflammatory factor (interleukin-10) to detect their levels in the liver and kidney of mice using immunohistochemistry. The results are shown in Fig. 9A through D. The experimental data revealed that the model group exhibited a significant inflammatory response compared to the blank control group, with elevated secretion levels of the pro-inflammatory cytokines IL-6 and TNF-α. Notably, after intervention with SH and SNH, a dose-dependent immunomodulatory effect was observed in the liver and kidney tissues of mice: the

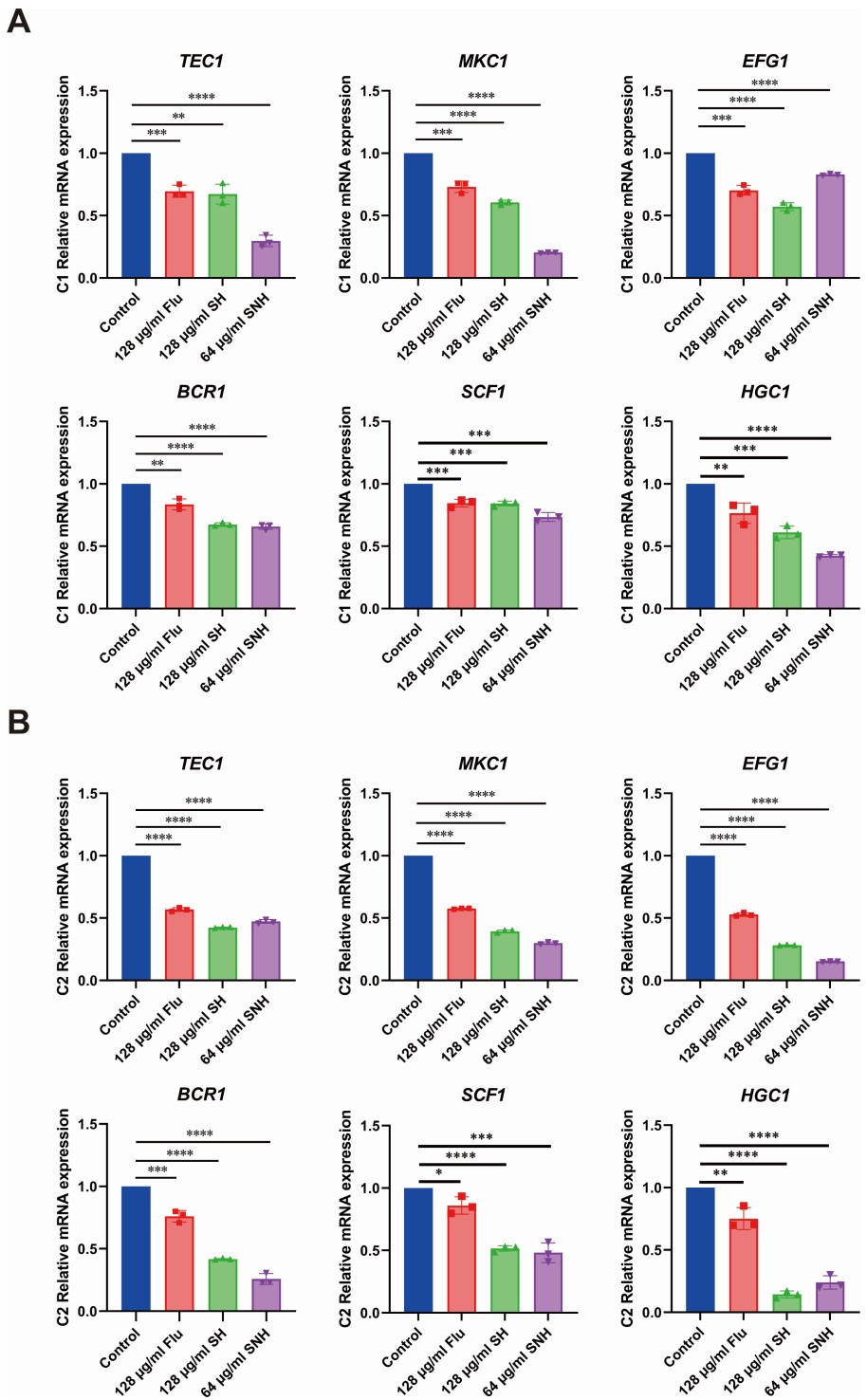

**FIG 6** SH and SNH inhibit the expression of virulence factor genes in *C. auris*. (A and B) After SH and SNH treatments, the mRNA expression levels of six virulence factors in *C. auris* strains C1 (A) and C2 (B).

secretion levels of IL-6 and TNF-α decreased compared to the model group, while the expression level of the anti-inflammatory factor IL-10 was significantly upregulated. The SNH group also showed a similar regulatory trend under treatment. Further mechanistic studies indicated that SH and SNH might effectively alleviate the cytokine storm triggered by *C. auris* infection by inhibiting the activation of the NF-κB signaling pathway

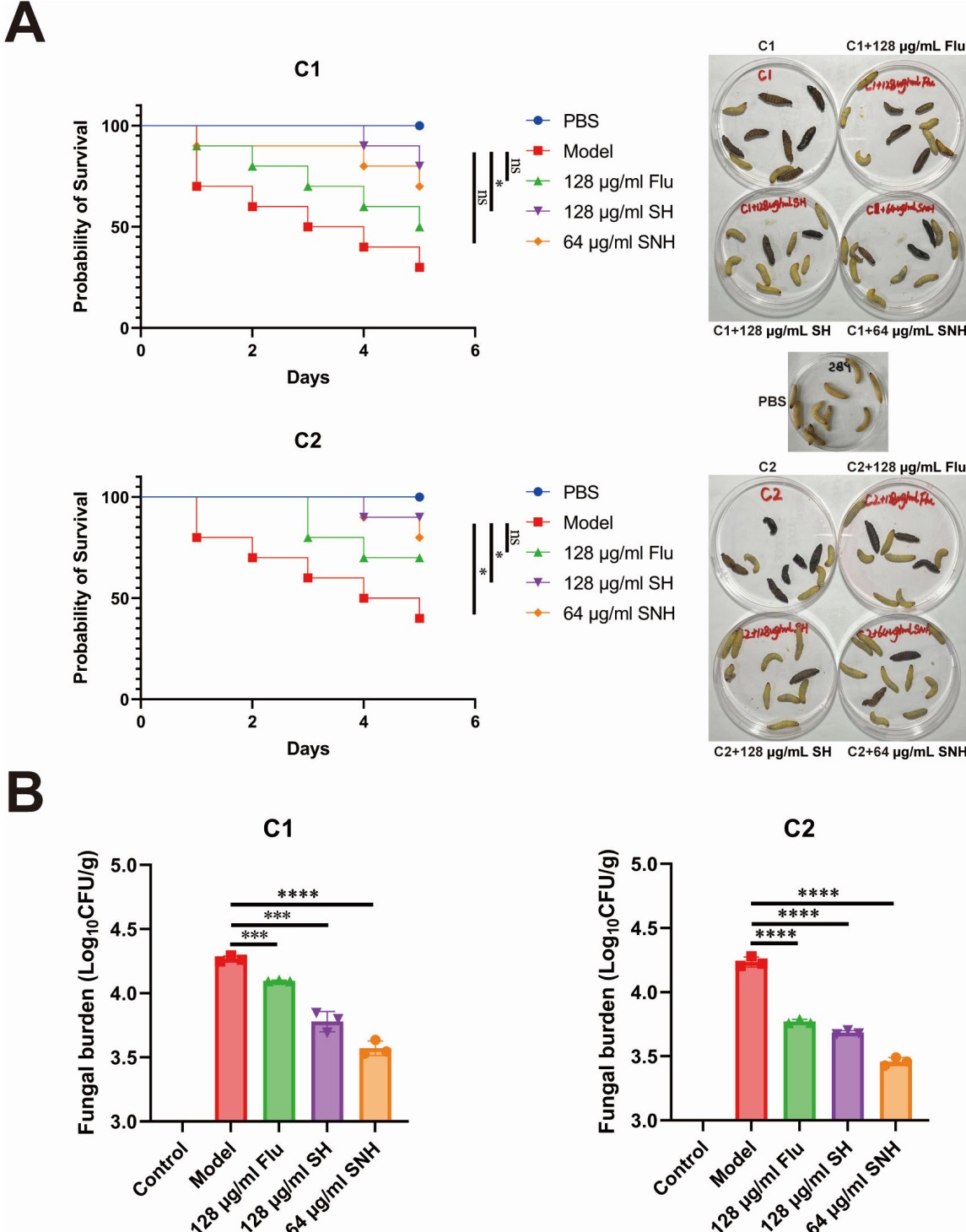

**FIG 7** SH and SNH inhibited the infection of *C. auris* in *G. mellonella* larvae. (A) The Kaplan-Meier method and log-rank test were utilized to analyze survival curves and identify differences. PBS represents the negative control group by injection of PBS. (B) Fungal load in *G. mellonella* larvae infected with C1 and C2.

and reducing the phosphorylation level of STAT3. These findings suggest that SH and SNH, as novel antifungal adjuvant therapies, can improve the excessive inflammatory response caused by invasive fungal infections through a multi-target regulatory mode.

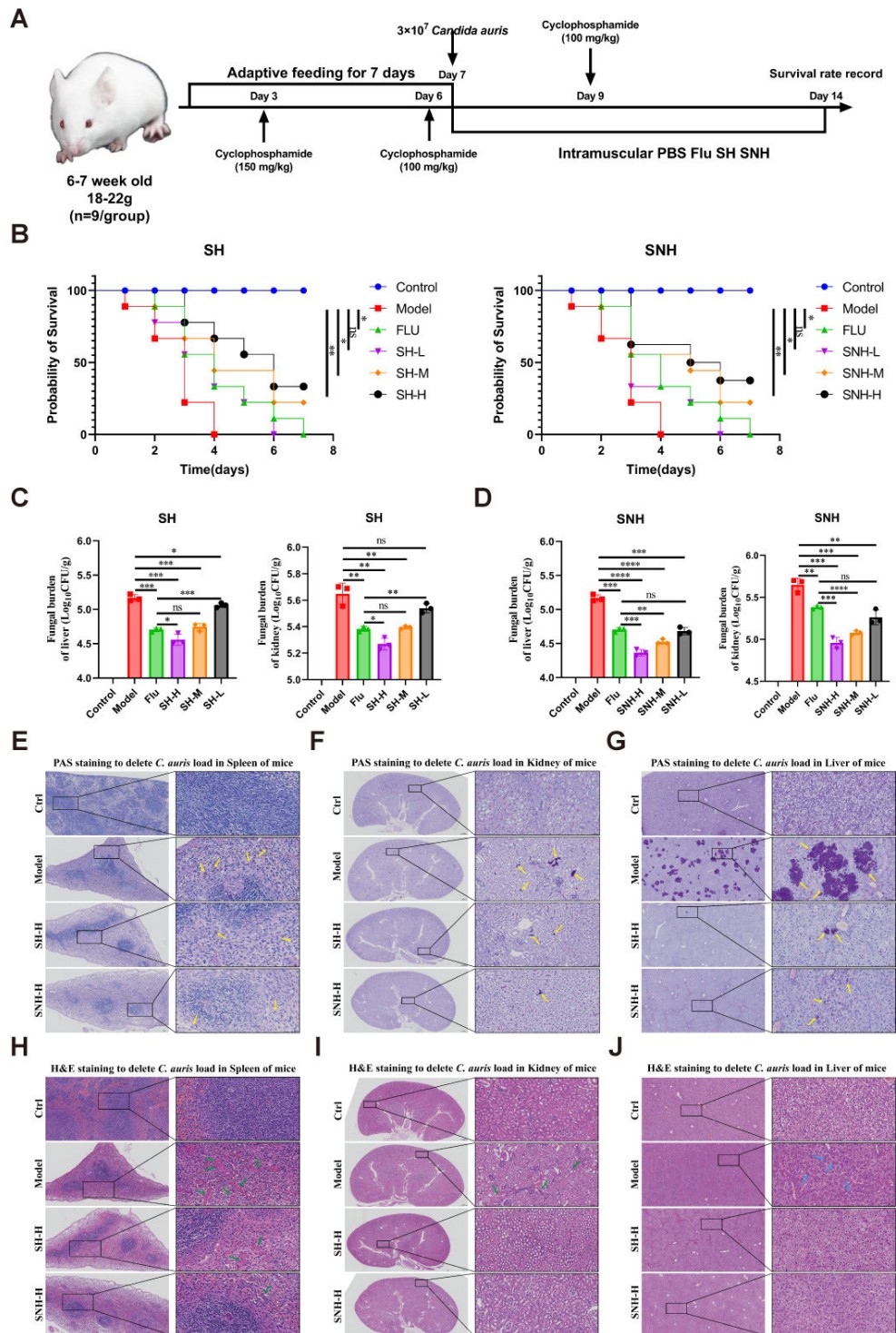

**FIG 8** SH and SNH inhibited the systemic infection of *C. auris* in mice. (A) Schematic diagram of mouse model for systemic *C. auris* infection. (B) Survival curve of mice infected with *C. auris*. (C and D) Fungal load in the liver and kidney of mice. (E–G) Representative PAS staining of spleen, kidneys, and liver from infected mice at 48 h post-infection. The yellow arrows indicated the *C. auris* cells in the tissue. (H–J) Representative images of hematoxylin and eosin (H&E) staining of mouse spleen, kidneys, and liver sections. The green arrows in the H&E staining indicate tissue damage, while the blue arrows indicate inflammatory cell infiltration. Note: L, M, and H mean low, medium, and high groups.

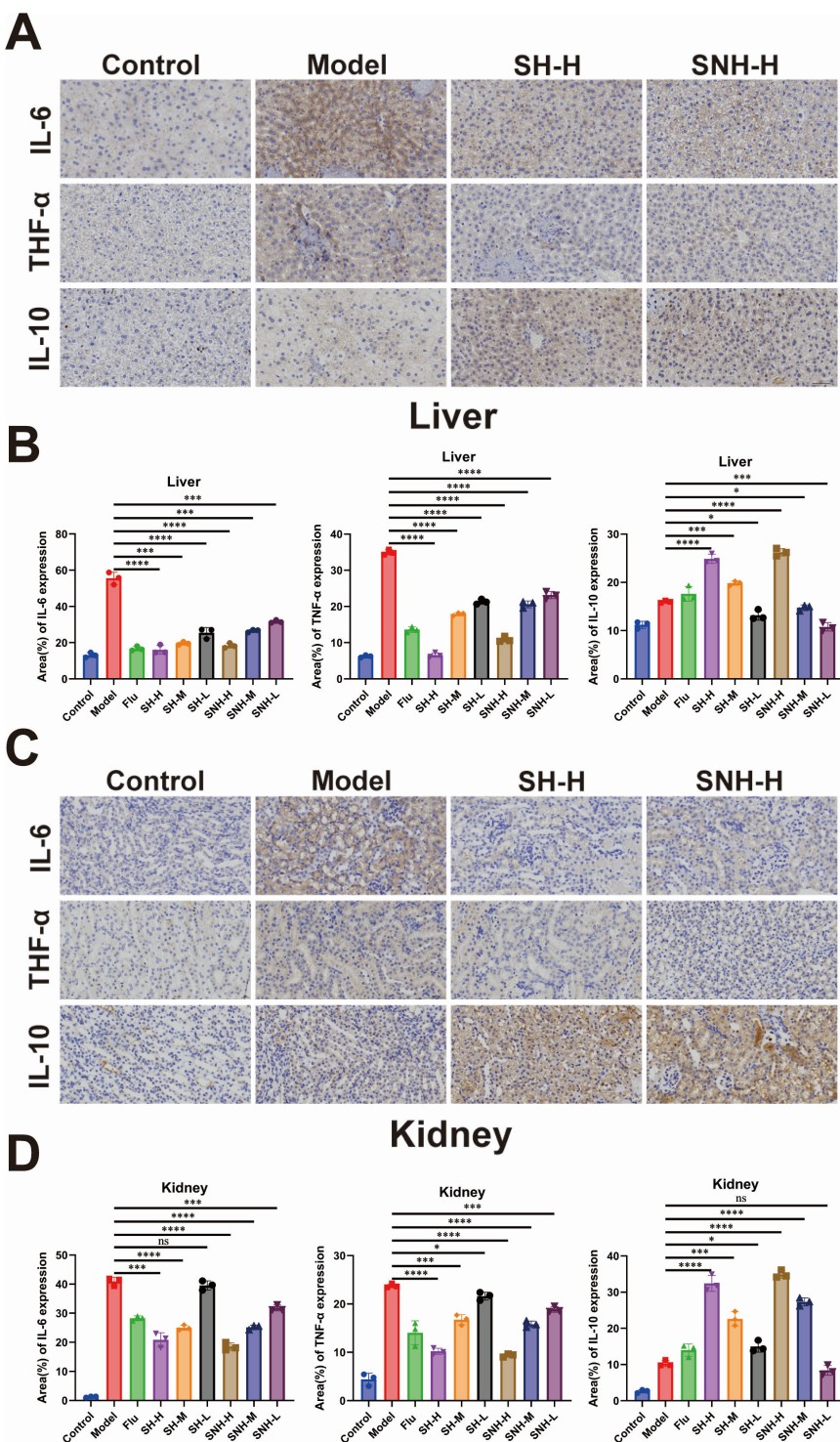

**FIG 9** SH and SNH reduced the inflammatory response to systemic infection of *C. auris* in mice. (A) Representative immunohistochemical images of mouse liver sections. Scale bar: 50 µm. (B) Quantitative immunohistochemical analysis of inflammatory factors in mouse liver tissue based on Image J software. (C) Representative immunohistochemical images of mouse kidney sections. Scale bar: 50 µm. (D) Quantitative immunohistochemical analysis of inflammatory factors in mouse kidney tissue based on Image J software.

## DISCUSSION

Recently, the annual proportion of non-*C. albicans* infections has been rising. All *Candida* pathogens, encompassing *C. albicans*, can cause superficial and systemic infections of varying degrees of severity (52). Furthermore, the most notable characteristic of *C. auris* is its resistance to drugs, including multidrug resistance, which results in treatment failures and poses significant challenges in controlling its spread. The antifungal drugs currently approved are categorized into four primary classes: polyenes, azoles, echinocandins, and the less frequently utilized flucytosine (53). Most significantly, clinical isolates from *C. auris* infections exhibit significant and occasionally untreatable resistance to all known antifungal classes, encompassing azoles, polyenes and echinocandins (54). Hence, the limited number of antifungal drug classes, coupled with the rising incidence of bloodstream infections and marked antifungal resistance, highlights the urgent requirement for more potent antifungal therapies. In our research, we found that SH and SNH exhibit strong growth inhibitory effects on *C. auris*. Compared to commonly used antifungal drugs such as fluconazole, SH and SNH have demonstrated even greater efficacy in inhibiting *C. auris*. Given the toxicity and high costs associated with currently available antifungal agents, SH and SNH exhibit promising potential as treatment options for *C. auris* infections.

Our research has revealed that SH and SNH exhibit potent inhibitory effects on the growth of *C. auris*. To delve into the antifungal mechanisms of these compounds against *C. auris*, we performed a comparative analysis of gene expression profiles between *C. auris* cells treated with SH and SNH and those left untreated. Our findings indicate that, in comparison to the untreated control group, the expression levels of 454 genes were significantly altered in the SH-treated group, whereas 532 genes underwent notable changes in the SNH-treated group. Microscopic observations further confirmed that SH and SNH effectively impair the early adhesion and aggregation processes of *C. auris*. In addition, our investigation into the impact of SH and SNH on *C. auris* biofilm formation showed that both compounds significantly influence this process. Intriguingly, we also observed a slight increase in the chitin and glucan content of the *C. auris* cell wall

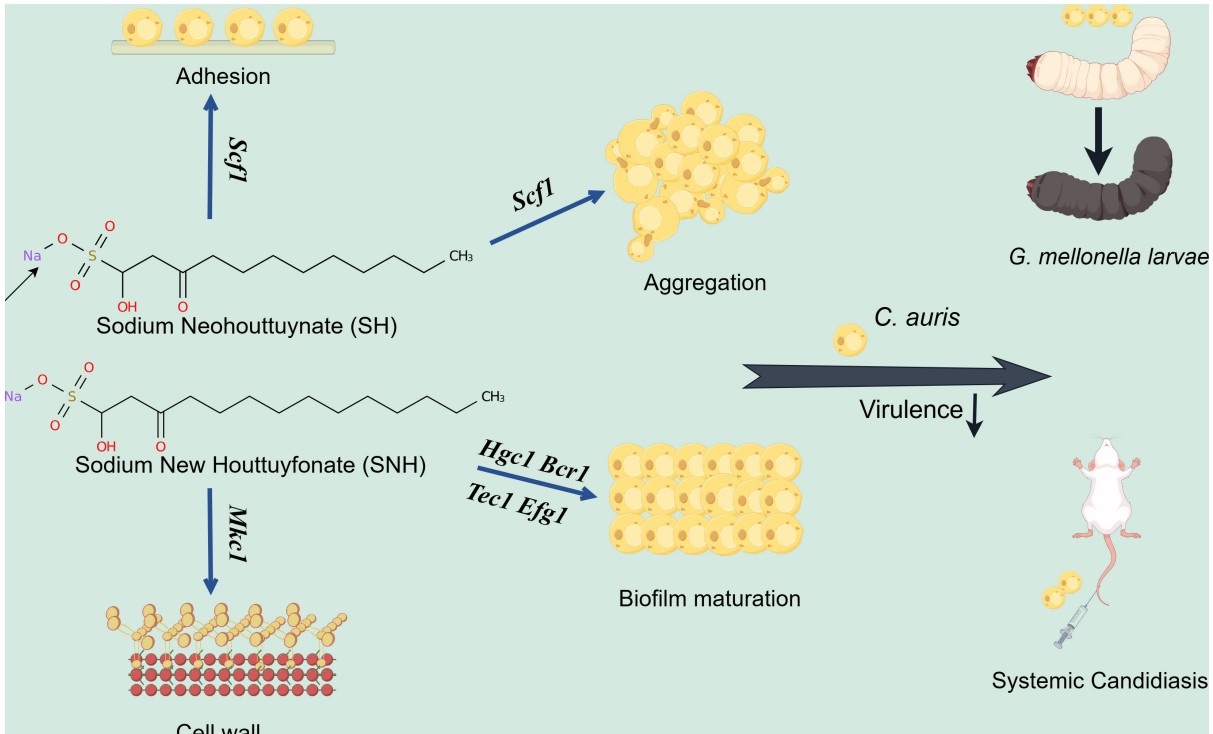

**FIG 10** Mechanism schematic representation of SH and SNH inhibits *C. auris*. The potent model image was drawn by figdraw (https://www.figdraw.com/).

upon treatment with SH and SNH. Further analysis of genes uniquely differentially expressed in the SH and SNH treatment groups revealed downregulation of multiple genes related to aggregation, adhesion, biofilm formation, cell wall remodeling, and other virulence factor genes, including *Tec1*, *Mkc1*, *Efg1*, *Bcr1*, *Scf1*, and *Hgc1*. Using *G. mellonella* larva infection models and systemic *C. auris* infection models in mice, our studies demonstrated that SH and SNH can markedly decrease the pathogenicity of *C. auris in vivo*. In summary, our experimental results suggest that SH and SNH inhibit the adhesion, aggregation, biofilm formation, and cell wall remodeling of *C. auris*, ultimately reducing its pathogenicity (Fig. 10).

*C. auris* has a propensity for adhering to and forming biofilms on indwelling medical devices such as intravascular catheters, which is a significant risk factor for systemic infections. A notable morphological characteristic of some isolates of *C. auris* is their ability to aggregate and form robust biofilms (55, 56). The *Scf1* gene is apparently unique to *C. auris*, and its expression mediates adhesion to both inert and biological surfaces across isolates from all five clades. Unlike typical fungal adhesins that function through hydrophobic interactions, *Scf1* relies on exposed cationic residues for surface association. *Scf1* is essential for the formation of *C. auris* biofilms, skin colonization, virulence in systemic infections, and colonization of medical devices (12). Our RT-qPCR results indicated a significant reduction in the expression level of *Scf1* after treatment with SH and SNH. In addition, SH and SNH can also significantly inhibit the early adhesion of *C. auris*. However, treatment with FLU did not yield satisfactory results. Fungal cell aggregation is regarded as the consequence of comprehensive conformational alterations on the fungal cell surface (57). In our research results, only a small amount of *C. auris* aggregated at the bottom of the Petri dishes after SH and SNH treatment, compared to the control group. Based on these findings, we conclude that SH and SNH can effectively inhibit the early adhesion and aggregation of *C. auris*.

Biofilms represent a primary form of microbial growth and play a crucial role in the development of clinical infections (44). As adhesion progresses, biofilm formation becomes a crucial factor in the virulence of *C. auris*. Biofilms exhibit higher resistance to antifungal drugs compared to planktonic cells. Regarding *C. auris*, virtually all isolates demonstrate resistance to the triazole drug FLU, with a significant proportion, nearly 40%, exhibiting a multidrug resistance phenotype (8, 58–60). However, the heightened resistance associated with biofilm formation poses an additional challenge to treatment. For example, echinocandin drugs are frequently prescribed for the management of invasive *C. auris* infections due to their relatively low resistance rates (61, 62). Yet, given the 2 to 512-fold increase in resistance observed in biofilms, these drugs are not anticipated to be efficacious in treating *C. auris* infections characterized by biofilm growth. The transcription factor *Bcr1* regulates the formation of fungal biofilms *in vivo* in catheter, denture, and vaginal models (63–65). Although *Bcr1* is not essential for hyphal morphogenesis, it functions as a positive regulator of hyphal-specific adhesins (65, 66). The G1 cyclin gene *Hgc1* contributes to the formation of biofilm, possibly by promoting filamentous formation during the proliferative phase (13). Recent studies have shown the roles of *Tec1* and *Ste12* transcription factors in regulating certain traits of *Candida* species, including biofilm formation and responses to low pH stress and high temperature (67). The transcription factor *Efg1* serves as a central transcriptional regulator of morphogenesis and was first characterized in *C. albicans* in 1997. *Efg1* can regulate filamentation, biofilm formation, white/opaque switching, intestinal colonization, and virulence in *Candida* species (68). Our RT-qPCR results indicated a significant reduction in the expression level of *Bcr1*, *Hgc1*, *Tec1*, and *Efg1* after treatment with SH and SNH. The findings from XTT and crystal violet assays revealed that treatments with SH and SNH significantly suppressed the metabolic activity and reduced the biomass of *C. auris* biofilms, whereas FLU treatment showed negligible inhibitory effects. CLSM imaging further revealed a dense packing of oval-shaped cells within the *C. auris* biofilms. In addition, SH and SNH treatments led to a decrease in both biomass and cell density of

the *C. auris* biofilms, whereas FLU treatment failed to exhibit any inhibitory impact on biofilm formation.

Currently, the virulence of *C. auris* is evaluated through both mammalian and non-mammalian models (69). The mouse model (70), a commonly used mammalian model, typically involves systemic, oral, and vaginal infection models, as well as skin and gastrointestinal colonization models. Non-mammalian models frequently employed include zebrafish *Danio rerio* larvae (71), the fruit fly *Drosophila melanogaster* (72), *G. mellonella* (73) larvae, and *Caenorhabditis elegans* (74). In our study, we utilized the *G. mellonella* larva model and a mouse systemic infection model to assess the virulence of *C. auris* following treatment with SH and SNH. Notably, the survival rates of *G. mellonella* larvae and mice in the groups treated with SH and SNH were significantly enhanced, and the fungal loads *in vivo* were markedly decreased compared to the model group. In addition, we conducted pathological examinations of the liver, kidney, and spleen in mice. The results indicated that the tissues of mice treated with high doses of SH and SNH exhibited less damage and only minimal inflammatory cell infiltration compared to the model group. IL-6 is a prototypical cytokine involved in maintaining homeostasis. When homeostasis is disrupted by infection or tissue damage, IL-6 is promptly produced and aids the host in counteracting this emergency stress by activating acute-phase and immune responses (75). IL-10 is a cytokine with pleiotropic immunosuppressive functions and is the founding member of the IL-10 family of cytokines (76). TNF-α is a cytokine that exerts pleiotropic effects on various cell types. It has been identified as a primary regulator of inflammatory responses and is known to be involved in the pathogenesis of several inflammatory and autoimmune diseases (77). Finally, we employed immunohistochemical techniques to quantitatively analyze the expression levels of inflammation-related cytokines in mouse liver and kidney tissues. The results demonstrated that, compared to the model control group, the SH and SNH intervention groups exhibited significant anti-inflammatory effects: the positive staining areas of IL-6 and TNF-α were markedly reduced, while the expression of the anti-inflammatory cytokine IL-10 was significantly upregulated. These findings indicate that SH and SNH can effectively alleviate the inflammatory cascade triggered by systemic *C. auris* infection by bidirectionally modulating the balance between pro-inflammatory and anti-inflammatory mediators.

The fungal cell wall is crucial for the survival of fungal pathogens, as its chemical composition determines the immune recognition of *C. auris* (78). Studies have demonstrated that SH, either alone or in combination with FLU, exhibits relatively strong antifungal potential against *Candida* species, with the underlying mechanism potentially related to the synthesis and transport of β-1,3-glucan (79). The CWI pathway plays a crucial role in the biogenesis of the cell wall in *C. albicans* and other fungal species, while the CWI pathway is mediated by the *Mkc1* gene (80). Our RT-qPCR results indicated a decrease in the expression level of *Mkc1* after treatment with SH and SNH. In addition, our research indicates that SH and SNH can slightly increase chitin deposition in *C. auris*, with a notable enhancement of chitin deposition observed in the region of newly emerged buds of *C. auris*. Meanwhile, they can also slightly elevate the expression of exposed β-1,3-glucan. We hypothesize that after treatment with SH and SNH, *C. auris* may regulate the spatial dynamic distribution of the chitin synthesis pathway by inducing a cell wall stress response. Conversely, treatment with FLU results in a mild increase in both chitin and β-1,3-glucan levels in *C. auris*. Consequently, there exist distinctions in the impact of SH and SNH on cell wall remodeling in drug-resistant strains of *C. auris*, compared to their effects on non-resistant strains.

Through the addition reaction between houttuynin and sodium bisulfite, SH is obtained, which retains the main pharmacological activity of houttuynin and has more stable physicochemical properties (20). SNH, an analog of SH, has two additional methylene groups in its fatty chain compared to SH. The pharmacological effects of SNH are basically the same as those of SH. SH can affect the gene expression of various bacteria and fungi, inhibit biofilm formation, alter the cellular morphology of bacteria or

fungi, and exhibit synergistic effects when combined with certain existing antibacterial or antifungal drugs. The combination of SH and EDTA-Na$_2$ can significantly enhance the growth inhibition and biofilm removal effects of SH on fungi (81). SH can serve as a sensitizer for conventional antifungal drugs in the treatment of oropharyngeal candidiasis (OPC), and it can also inhibit filamentation in dual biofilms of *C. albicans* and *Candida glabrata*. Studies have shown that SNH can inhibit the initial adhesion of fungi and prevent the transition of *C. albicans* from the yeast phase to the hyphal phase (29). Here, we demonstrate that houttuynin derivatives, SH and SNH, can effectively repress the adhesion, aggregation, and biofilm formation of *C. auris*. Unlike the significant effects of SH on β-1,3-glucan exposure on *C. albicans*, we only discovered that SH and SNH can mildly increase the β-1,3-glucan exposure of *C. auris*. These results implied that the antimicrobial mechanism of SH and SNH against *C. auris* is not identical to the mechanism of SH and SNH against *C. albicans* and is worth further investigation.

However, the results of this study are not without their limitations. Notably, the MIC values of SH and SNH against *C. auris*, ranging from 64 to 128 µg/mL, are considered relatively high. Furthermore, our research did not identify the specific targets of SH and SNH against *C. auris*. To address these shortcomings, our research team is currently engaged in further in-depth studies. In addition, we envision the potential for further structural modifications of SH and SNH to bolster their efficacy against drug-resistant *C. auris*.

In summary, this study indicates that SH and SNH have the potential to be effective antifungal and anti-biofilm agents. Through *in vitro* phenotypic analysis and *in vivo* infection analysis, we have confirmed their inhibitory effect on multidrug-resistant *C. auris*. Further research is needed to elucidate the specific mechanisms by which SH and SNH act against *C. auris* and to evaluate their potential clinical applications in the pharmaceutical industry.

## Lay summary

SH and SNH exhibit potent inhibitory effects on the growth, adhesion, aggregation, and biofilm formation of *C. auris*, thus mitigating its colonization and pathological impact on the host.

## ACKNOWLEDGMENTS

The authors are grateful for the helpful and constructive suggestions of Dr. Guanghua Huang at Fudan University. The authors express their gratitude for the assistance provided by the public experimental research center at the College of Integrated Chinese and Western Medicine, Anhui University of Chinese Medicine.

This work was sponsored by the projects funded by the National Natural Science Foundation of China (grant number: 82374121), the Natural Science Foundation (Key project) of the University in Anhui Province (grant number: 2023AH050727, 2023AH04011, 2024AH050921), Hefei Municipal Natural Science Foundation (grant number: HZR2436), and Anhui Postdoctoral Scientific Research Program Foundation (grant number: 2024 A755). The funders had no role in research design, data collection and analysis, decision to publish, or preparation of the manuscript.

G.Y. and R.Y. performed the experiments. G.Y., R.Y., X.Z., and Q.X. contributed to data analysis. X.N. was involved in providing the resources. T.W. and D.W. contributed to conception and design. G.Y., C.M., W.W., and T.W. drafted the manuscript and critically revised the manuscript. D.W. supervised the project. All authors reviewed the manuscript and approved the final manuscript.

## AUTHOR AFFILIATIONS

[1]Department of Pathogenic Biology and Immunology, College of Integrated Chinese and Western Medicine, Anhui University of Chinese Medicine, Hefei, Anhui, China

[2]Key laboratory of Xin'an Medicine, Ministry of Education, Research Institute of Integrated Traditional Chinese and Western Medicine, Anhui Academy of Chinese Medicine, Hefei, Anhui, China

## AUTHOR ORCIDs

Tianming Wang http://orcid.org/0000-0001-5065-4553
Daqiang Wu http://orcid.org/0000-0003-1581-1538

## FUNDING

| Funder | Grant(s) | Author(s) |
| --- | --- | --- |
| National Natural Science Foundation of China | 82374121 | Tianming Wang |
| Natrual Science foundation of the university of Anhui Province | 2023AH050727, 2023AH04011, 2024AH050921 | Daqiang Wu |
| Hefei Municipal Natural Science Foundation | HZR2436 | Daqiang Wu |
| Anhui Postdoctoral Science Foundation | 2024A755 | Daqiang Wu |

## AUTHOR CONTRIBUTIONS

Guangyuan Yang, Formal analysis, Investigation, Methodology, Writing – original draft | Ruotong Yang, Investigation, Methodology | Xiaoxiao Zhu, Data curation, Investigation, Methodology | Qianwen Xu, Investigation, Methodology | Xiaojia Niu, Methodology, Resources | Chengui Miao, Conceptualization | Wenfan Wei, Data curation | Changzhong Wang, Resources | Tianming Wang, Funding acquisition, Project administration, Validation, Writing – review and editing.

## DATA AVAILABILITY

The study includes the original findings, which are incorporated in the Materials and Methods section. For additional inquiries, please contact the corresponding authors.

## ETHICS APPROVAL

The animal experiment had been approved and supervised through the Animal Ethical Committee of the Anhui University of Chinese Medicine (Animal Ethics Approval Number: AHUCM-mouse-2025008).

## ADDITIONAL FILES

The following material is available online.

### Open Peer Review

**PEER REVIEW HISTORY (review-history.pdf).** An accounting of the reviewer comments and feedback.

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
