## [Reviewer comments · Microbiology Spectrum]

Microbiology Spectrum

In vitro and in vivo activity of sodium houttuynate and sodium new houttuynate against *Candida auris* infection by affecting adhesion, aggregation, and biofilm formation abilities

Guangyuan Yang, Ruotong Yang, Xiaoxiao Zhu, qianwen Xu, Xiaojia Niu, Chengui Miao, Wenfan Wei, Changzhong Wang, Tianming Wang, and Da-qiang Wu

Corresponding Author(s): Da-qiang Wu, Anhui University of Chinese Medicine

Review Timeline:

Submission Date:	January 23, 2025
Editorial Decision:	February 17, 2025
Revision Received:	April 18, 2025
Accepted:	May 26, 2025

Editor: Gustavo Goldman

Reviewer(s): The reviewers have opted to remain anonymous.

Transaction Report:

DOI: <https://doi.org/10.1128/spectrum.00222-25>

Re: Spectrum00222-25 (In vitro and in vivo activity of sodium houttuynonate and sodium new houttuynonate against *Candida auris* infection by affecting adhesion, aggregation, and biofilm formation abilities)

Dear Dr. Da-qiang Wu:

Thank you for the privilege of reviewing your work. Below you will find my comments, instructions from the Spectrum editorial office, and the reviewer comments.

Your manuscript has been reviewed by two reviewers who provided several comments and suggestions aiming to improve it. Please, submit a revised version together with a rebuttal letter addressing point-by-point raised by each reviewer.

Revision Guidelines

Sincerely,
Gustavo H. Goldman
Editor
Microbiology Spectrum

Reviewer #1 (Comments for the Author):

Review Spectrum00222-25

Candida auris is a major cause of invasive infections in both acute and long-term healthcare settings. The fungus has been

reported in over 47 countries and can colonize skin and mucosal surfaces, forming biofilms. It is resistant to certain standard disinfectants and is a multidrug-resistant organism linked to hospital outbreaks with high mortality rates. Due to its resistance to many antifungal agents including azole, echinocandins and polyenes, it becomes difficult to treat infections. Hence, there is an urgent need for more effective agents that will be useful in treating *C. auris* infections. In this study, the authors conducted *in vitro* and *in vivo* assays to explore the potential of SH and SNH in inhibiting the growth, adhesion, aggregation, and biofilm formation of *Candida auris*. Below are my comments:

Major Comments:

1. Confirmation of Strains (Materials and Methods):

- o The authors should provide details on how the *C. auris* strains were reconfirmed in their laboratory before determining the Minimum Inhibitory Concentration (MIC). This ensures that the strains used are correctly identified.

2. Clade Information:

- o Since *C. auris* is classified into six clades, it's critical to mention which clades these strains belong to. This would make it more specific and relevant to understand the potency of SH and SNH against different strains of *C. auris*.

- o E.g. the aggregating and non-aggregating cells of *C. auris* differ in several aspects, including their ability to form biofilms, as well as their virulence and resistance to antifungal drugs.

3. Antifungal Susceptibility Testing (AFST):

- o Provide more specific details on the drugs tested against the six strains, including MIC ranges, the guidelines followed when reading the MIC (e.g., CLSI guidelines/CDC recommendations?), and what controls were included during AFST. CLSI guidelines recommend including *C. parapsilosis* 22019 and *C. krusei* 6258 ATCC strains as controls.

4. Strain Usage and Selection:

- o It seems that different strains were used for different experiments. The authors should specify which strain was used in each experiment and provide a rationale for why certain strains were selected for the particular experiment.

- o Is there a reason why you selected to use 128 µg/ml of fluconazole instead of 1024 µg/ml? and why was the echinocandins not tested especially since they are used for the treatment of *C. auris* infection.

5. Strain Selection:

- o Is there a specific reason why 12373 and 12767 strains were not tested in all the experiments, especially considering that strain 12373 had lower MIC values compared to the other strains. It would be valuable to see the results from this strain in various assays.

6. Methodology Details:

- o For each experiment, it's important to specify the protocols followed (e.g., XTT assays should mention the specific reference or protocol used). This adds clarity and helps other researchers replicate the study.

7. Larvae Experiment Clarification:

- o There is a discrepancy in the larvae experiment where 90 larvae were used, but only 80 were reported in the results section. Clarify what happened to the 10 unreported larvae. Also, it's necessary to indicate which strains were tested and why only those strains were selected out of the six.

8. Interpretation of Results:

- o The results suggest that higher doses of SH and SNH are required for efficacy against *C. auris*. It would be helpful to elaborate on this point: does this indicate limited potency or is it reflective of how these agents function *in vivo*? Furthermore, does this depend on the type of strain tested?

9. Comparison of Doses:

- o For the mouse model, is there a reason why only high doses of SH and SNH were used in contrast to fluconazole, which had a much lower dose.

Minor Comments:

1. Line 99:

- o Amphotericin B is not a class; it is an individual antifungal drug. This needs correction.

2. Line 176:

- o Specify which drugs were tested in this section to make the methodology clearer.

3. Line 226:

- o Indicate which strains were tested for, this should be done for all experiments.

4. Line 240:

- o Correct "fun-gal" to "fungal."

5. Lines 241-274:

- o Ensure there is no unnecessary repetition between lines 241-244 and 244-246.

6. Line 251-252:

- o Check if this should read "fungal suspension" instead of any bacterial suspension.

7. Line 380:

- o Specify which drugs the strains are resistant to.

8. Line 602:

- o Again, Amphotericin B is not a class; it is an individual antifungal drug. This needs correction.

9. Line 606-607:

- o The term "commonly mentioned antifungal drugs" is too vague-are you referring to fluconazole and amphotericin B? This statement needs clarification to avoid confusion.

10. Line 612:

o Did you mean "antifungal mechanisms".

11. Figure 6:

o Rephrase the labels in Figure 6 (e.g., "A = C1?" and "B = C2?").

Reviewer #2 (Comments for the Author):

See attached review.

Review Spectrum00222-25

Candida auris is a major cause of invasive infections in both acute and long-term healthcare settings. The fungus has been reported in over 47 countries and can colonize skin and mucosal surfaces, forming biofilms. It is resistant to certain standard disinfectants and is a multidrug-resistant organism linked to hospital outbreaks with high mortality rates. Due to its resistance to many antifungal agents including azole, echinocandins and polyenes, it becomes difficult to treat infections. Hence, there is an urgent need for more effective agents that will be useful in treating *C. auris* infections. In this study, the authors conducted in vitro and in vivo assays to explore the potential of SH and SNH in inhibiting the growth, adhesion, aggregation, and biofilm formation of *Candida auris*. Below are my comments:

Major Comments:

1. **Confirmation of Strains (Materials and Methods):**
 - The authors should provide details on how the *C. auris* strains were reconfirmed in their laboratory before determining the Minimum Inhibitory Concentration (MIC). This ensures that the strains used are correctly identified.
2. **Clade Information:**
 - Since *C. auris* is classified into six clades, it's critical to mention which clades these strains belong to. This would make it more specific and relevant to understand the potency of SH and SNH against different strains of *C. auris*.
 - E.g. the aggregating and non-aggregating cells of *C. auris* differ in several aspects, including their ability to form biofilms, as well as their virulence and resistance to antifungal drugs.
3. **Antifungal Susceptibility Testing (AFST):**
 - Provide more specific details on the drugs tested against the six strains, including MIC ranges, the guidelines followed when reading the MIC (e.g., CLSI guidelines/CDC recommendations?), and what controls were included during AFST. CLSI guidelines recommend including *C. parapsilosis* 22019 and *C. krusei* 6258 ATCC strains as controls.
4. **Strain Usage and Selection:**
 - It seems that different strains were used for different experiments. The authors should specify which strain was used in each experiment and provide a rationale for why certain strains were selected for the particular experiment.
 - Is there a reason why you selected to use 128 µg/ml of fluconazole instead of 1024µg/ml? and why was the echinocandins not tested especially since they are used for the treatment of *C. auris* infection.
5. **Strain Selection:**
 - Is there a specific reason why 12373 and 12767 strains were not tested in all the experiments, especially considering that strain 12373 had lower MIC values compared to the other strains. It would be valuable to see the results from this strain in various assays.
6. **Methodology Details:**
 - For each experiment, it's important to specify the protocols followed (e.g., XTT assays should mention the specific reference or protocol used). This adds clarity and helps other researchers replicate the study.
7. **Larvae Experiment Clarification:**

- There is a discrepancy in the larvae experiment where 90 larvae were used, but only 80 were reported in the results section. Clarify what happened to the 10 unreported larvae. Also, it's necessary to indicate which strains were tested and why only those strains were selected out of the six.
8. **Interpretation of Results:**
 - The results suggest that higher doses of SH and SNH are required for efficacy against *C. auris*. It would be helpful to elaborate on this point: does this indicate limited potency or is it reflective of how these agents function in vivo? Furthermore, does this depend on the type of strain tested?
 9. **Comparison of Doses:**
 - For the mouse model, is there a reason why only high doses of SH and SNH were used in contrast to fluconazole, which had a much lower dose.

Minor Comments:

1. **Line 99:**
 - Amphotericin B is not a class; it is an individual antifungal drug. This needs correction.
2. **Line 176:**
 - Specify which drugs were tested in this section to make the methodology clearer.
3. **Line 226:**
 - Indicate which strains were tested for, this should be done for all experiments.
4. **Line 240:**
 - Correct "fun-gal" to "fungal."
5. **Lines 241–274:**
 - Ensure there is no unnecessary repetition between lines 241-244 and 244-246.
6. **Line 251-252:**
 - Check if this should read "fungal suspension" instead of any bacterial suspension.
7. **Line 380:**
 - Specify which drugs the strains are resistant to.
8. **Line 602:**
 - Again, Amphotericin B is not a class; it is an individual antifungal drug. This needs correction.
9. **Line 606-607:**
 - The term "commonly mentioned antifungal drugs" is too vague—are you referring to fluconazole and amphotericin B? This statement needs clarification to avoid confusion.
10. **Line 612:**
 - Did you mean "antifungal mechanisms".
11. **Figure 6:**
 - Rephrase the labels in Figure 6 (e.g., "A = C1?" and "B = C2?").

Title:

Sodium Houttuynonate and Sodium New Houttuynonate Inhibit Growth, Adhesion, and Biofilm Formation of *Candida auris*

Summary:

This study explores the antifungal potential of sodium houttuynonate (SH) and sodium new houttuynonate (SNH) against *Candida auris*, a multidrug-resistant fungal pathogen. The authors use in vitro and in vivo models to evaluate the effects of SH and SNH on *C. auris* growth, adhesion, aggregation, biofilm formation, and systemic infection.

Key findings include:

- **Growth Inhibition:** SH and SNH effectively inhibit the growth of fluconazole-resistant *C. auris* strains but do not exhibit strong fungicidal activity.
- **Virulence Gene Suppression:** Transcriptomic analysis and RT-qPCR demonstrate that SH and SNH downregulate key virulence-related genes (*Bcr1*, *Tec1*, *Mkc1*, *Scf1*, *Hgc1*, *Efg1*), which are involved in adhesion, aggregation, biofilm formation, and cell wall remodeling.
- **Adhesion and Aggregation:** SH and SNH impair *C. auris*' ability to adhere to surfaces and form aggregations, reducing colonization potential.
- **Biofilm Disruption:** XTT assays, crystal violet staining, and confocal laser scanning microscopy reveal that SH and SNH significantly inhibit *C. auris* biofilm formation, a key factor in antifungal resistance.
- **Cell Wall Modifications:** SH and SNH mildly increase chitin deposition and expose β -1,3-glucan in *C. auris*, suggesting structural changes in the fungal cell wall that may affect immune recognition and drug sensitivity.
- **In Vivo Efficacy:** Both *Galleria mellonella* larvae and murine systemic candidiasis models demonstrate that SH and SNH reduce fungal load and host tissue damage, improving survival rates.

Strengths:

- **Relevance and Novelty:** The study addresses a critical medical challenge—drug resistance in *C. auris*—and evaluates novel antifungal candidates derived from traditional Chinese medicine.
- **Comprehensive Experimental Design:** The combination of in vitro assays, transcriptomic analysis, and in vivo infection models strengthens the reliability of the findings.
- **Mechanistic Insights:** The investigation into gene expression changes and cell wall remodeling provides valuable insights into how SH and SNH affect fungal physiology.
- **Potential for Clinical Application:** The findings highlight SH and SNH as promising antifungal agents with potential applications in treating *C. auris* infections, particularly those resistant to conventional therapies.

Suggestions for Improvement:

- Line 252- XTT assay, did you mean fungal not bacterial, see sentence “The XTT assay was employed to assess the ability of the drug to inhibit *C. auris* biofilms. The concentrations of each bacterial suspension were adjusted to 2×10^3 cells/mL using RPMI-1640 medium. 100 μ L of each bacterial suspension was added to a 96-well plate. After static incubation ...”
- Line 503, typo oof instead of of.

Mechanism of Action on Cell Wall Components

- The study shows increased chitin deposition and β -1,3-glucan exposure but does not fully elucidate how these changes affect *C. auris*' drug susceptibility and/ or immune evasion. Has the group looked at extracellular vesicle trafficking or perhaps they can do quick staining like Dil stains to see if increased chitin is mostly around new buds, is it shedding it, how could it be protective to *C. auris* growth. Look at a similar paper by De Jesus et al. Capsular localization of the *Cryptococcus neoformans* polysaccharide component galactoxylomannan. *Eukaryot Cell*. 2009 Jan;8(1):96-103, that highlights changes in the fungal bud.

Host Immune Response in In Vivo Models

- While *G. mellonella* and murine models demonstrate reduced fungal burden and tissue damage, the study does not assess host immune responses. Analyzing cytokine levels or immune cell recruitment in infected tissues could provide some at a glance insights into how SH and SNH modulate host-pathogen interactions.

Overall Evaluation:

This study presents compelling evidence for the antifungal potential of SH and SNH against *C. auris*, with promising implications for clinical applications. The research is well-structured, employs rigorous methodologies, and provides mechanistic insights into how SH and SNH impair *C. auris* virulence. However, additional mechanistic studies and comparative efficacy analyses could further strengthen the manuscript's impact.

Recommendation:

- **Accept with Revisions**

Point to point response

Dear editor and reviewers ,

We are deeply grateful for the comments and constructive suggestions on our manuscript of editor and reviewers. The point-to-point responses are as follows:

Reviewer #1:

Candida auris is a major cause of invasive infections in both acute and long-term healthcare settings. The fungus has been reported in over 47 countries and can colonize skin and mucosal surfaces, forming biofilms. It is resistant to certain standard disinfectants and is a multidrug-resistant organism linked to hospital outbreaks with high mortality rates. Due to its resistance to many antifungal agents including azole, echinocandins and polyenes, it becomes difficult to treat infections. Hence, there is an urgent need for more effective agents that will be useful in treating *C. auris* infections. In this study, the authors conducted in vitro and in vivo assays to explore the potential of SH and SNH in inhibiting the growth, adhesion, aggregation, and biofilm formation of *Candida auris*.

Response: We are deeply grateful for the positive comments on our manuscript of reviewer.

Major Comments:

1.Confirmation of Strains (Materials and Methods):

The authors should provide details on how the *C. auris* strains were reconfirmed in their laboratory before determining the Minimum Inhibitory Concentration (MIC). This ensures that the strains used are correctly identified.

Response: We are deeply grateful for the comment on our manuscript of reviewer. Genomic DNA was extracted from six clinical isolates of *C. auris*, and the ITS region was amplified using specific primers (ITS1/ITS4). The amplified products were sequenced and compared using BLAST. Furthermore, a phylogenetic tree was constructed based on the maximum likelihood method (MEGA 12, Bootstrap=1000) to confirm that all strains belonged to *C. auris*, providing a reliable basis for subsequent research (see Fig R1 for detailed data). The contents have been added in the revised manuscript as follows: “In this study, the internal transcribed spacer (ITS) sequences of six fluconazole resistant *C. auris* strains, namely C1, C2, C3, C4, 12373, and 12767, were obtained through PCR amplification, and a phylogenetic tree was constructed based on the ITS sequences. The molecular identification results (data not shown) showed that six clinically isolated strains were confirmed as *C. auris*.”

Figure R1. (A) Analysis results of agarose gel electrophoresis for the amplified products of the internal transcribed spacer (ITS) sequences from different isolates of *C. auris*. (B) Phylogenetic tree of *C. auris* constructed based on ITS sequences.

2. Clade Information:

Since *C. auris* is classified into six clades, it's critical to mention which clades these strains belong to. This would make it more specific and relevant to understand the potency of SH and SNH against different strains of *C. auris*.

E.g. the aggregating and non-aggregating cells of *C. auris* differ in several aspects, including their ability to form biofilms, as well as their virulence and resistance to antifungal drugs.

Response: We are deeply grateful for the comment on our manuscript of reviewer. Through drug-resistant phenotypic identification and collaborative communication with Professor Huang Guanghua's team at Fudan University who provide the *C. auris* strains, and phenotypic identification of fluconazole resistance, we have determined that the aforementioned strains all belong to the South Asian clade (Clade I) of *C. auris*. The contents have been added in the revised manuscript as follows: "Phylogenetic analysis, in combination with collaborative identification by Professor Huang Guanghua's team at Fudan University, indicated that the aforementioned strains all belonged to the South Asia Clade (Clade I) of *C. auris*."

3. Antifungal Susceptibility Testing (AFST):

Provide more specific details on the drugs tested against the six strains, including MIC ranges, the guidelines followed when reading the MIC (e.g., CLSI guidelines/CDC recommendations?), and what controls were included during AFST. CLSI guidelines recommend including *C. parapsilosis* 22019 and *C. krusei* 6258 ATCC strains as controls.

Response: We are deeply grateful for the comments on our manuscript of reviewer. According to the Clinical and Laboratory Standards Institute (CLSI) M27-A3 microbroth dilution method standards, this study utilized *Candida parapsilosis* ATCC 22019 as the quality control strain and strictly adhered to the experimental conditions of 24-hour incubation for caspofungin and amphotericin B (AMB). The test results revealed that the minimum inhibitory concentrations (MICs) of AMB and caspofungin against this quality control strain were both 0.5 µg/mL, which strictly fell within the expected ranges specified by the CLSI M27-A3 standards (AMB: 0.25–2 µg/mL; caspofungin: 0.25–1 µg/mL). These results validated the standardization of the antimicrobial susceptibility testing system, confirming that the media preparation, inoculation

concentration (2×10^3 CFU/mL), and endpoint interpretation method (100% visual growth inhibition) during the experiment all met the quality control requirements, thereby providing experimental evidence for the reliability of subsequent *C. auris* susceptibility data (see Table 3 for detailed data). We added this content in the material and methods section of revised manuscript as follows: “According to the CLSI M27-M44S protocol established by the Clinical and Laboratory Standards Institute (CLSI), the broth microdilution method in 96-well plates was used to determine the MIC of six *C. auris* strains against five drugs: SH, SNH, Fluconazole, Amphotericin B and Caspofungin. And a control was conducted using the *C. Parapsilosis* 22019 strain following the CLSI requirement.”

Strain	AmB	CAS
C. parapsilosis 22019	0.5 µg/mL	0.5 µg/mL

Note: AmB means Amphotericin B, and CAS means Caspofungin.

4. Strain Usage and Selection:

It seems that different strains were used for different experiments. The authors should specify which strain was used in each experiment and provide a rationale for why certain strains were selected for the particular experiment.

Is there a reason why you selected to use 128 µg/ml of fluconazole instead of 1024 µg/ml? and why was the echinocandins not tested especially since they are used for the treatment of *C. auris* infection.

Response: We are deeply grateful for the comments on our manuscript of reviewer. In the Materials and Methods section of this study, we have supplemented and labeled the information on the strains used in each experiment. We have also added all the six strains to determine the effects of SN and SNH on the growth curve, adhesion, aggregation and biofilm formation in the revised manuscript, and found that the similar phenotypic effects of SH and SNH on all these strains (Figure 1EF, Figure 3, Figure 4 attached as follows). Because, in this research, we have two test drugs and six tested *C. auris* strains, we did not have enough resources to perform the transcriptome, CWI and two in vivo infection model on all the six strains. If so, these results also will make the manuscript is too long to read. At present form, the manuscript contains 12238 words and 10 figures. Thus, we chose C1 as a typical fluconazole resistance *C. auris* strain to perform these assays.

Specifically, the test concentration of fluconazole (FLU) was uniformly set at 128 µg/mL to maintain consistency with previous preliminary experiments. Because, in this manuscript, the selected *C. auris* strains are resistant to fluconazole. Thus, we compare the antifungal effects of SH and SNH with fluconazole against *C. auris*. Thanks for the constructive advice of reviewer. We will compare the antifungal effects of SH and SNH with echinocandins such as caspofungin in our future research.

In addition, given that echinocandins (such as caspofungin) are the first-line treatment for *C. auris* infections, we subsequently conducted minimum inhibitory concentration (MIC) testing of caspofungin against the six *C. auris* strains using the CLSI M27 standard microbroth dilution method. The results showed that these *C. auris* strains are still susceptible to caspofungin, and the detailed susceptibility data are presented in Table 3.

Table 3 MIC of SH, SNH, Flu, AmB and CAS against *C. auris*

Strain	SH	SNH	Flu	AmB	CAS
C1	128 µg/mL	64 µg/mL	1024 µg/mL	2 µg/mL	4 µg/mL
C2	128 µg/mL	64 µg/mL	1024 µg/mL	1 µg/mL	4 µg/mL
C3	128 µg/mL	64 µg/mL	1024 µg/mL	1 µg/mL	4 µg/mL
C4	128 µg/mL	64 µg/mL	1024 µg/mL	1 µg/mL	4 µg/mL
12373	64 µg/mL	32 µg/mL	128 µg/mL	0.25 µg/mL	0.5 µg/mL
12767	64 µg/mL	32 µg/mL	128 µg/mL	2 µg/mL	0.5 µg/mL
C. parapsilosis 22019				0.5 µg/mL	0.5 µg/mL

Note: Flu means Fluconazole, AmB means Amphotericin B, and CAS means Caspofungin.

5. Strain Selection:

Is there a specific reason why 12373 and 12767 strains were not tested in all the experiments, especially considering that strain 12373 had lower MIC values compared to the other strains. It would be valuable to see the results from this strain in various assays.

Response: We are deeply grateful for the comments on our manuscript of reviewer. In this study, we systematically compared the biological characteristics of different *C. auris* strains by conducting a series of experiments, including growth dynamic analysis related to antimicrobial susceptibility (see Fig 1E-F for detailed data), early adhesion ability detection (see Fig 3A-B for detailed data), cell aggregation assays (see Fig 3C for detailed data), and in vitro biofilm formation ability evaluation (see Fig 4A-C for detailed data). The experimental results indicated that, compared to the other four clinically isolated strains, strains 12373 and 12767 exhibited significant disadvantages in key pathogenicity indicators such as growth rate and biofilm formation ability, suggesting that they may possess unique virulence regulatory mechanisms or defects in environmental adaptation.

Figure 1: SH and SNH exhibit significant inhibitory effects on the growth of *C. auris*.

(E-F) Growth inhibition curves of SH and SNH against two drug-resistant *C. auris* strains. Note: Flu means fluconazole.

Figure 3 SH and SNH inhibit the adhesion and aggregation of *C. auris*

(A) Images represent the state of *C. auris* adhesion at 3 h. Scale bars 25 µm. (B) Quantification of *C. auris* number by Image J. (C) Images represent the aggregation of *C. auris* following treatment with SH and SNH. Scale bars 5 µm.

Figure 4 SH and SNH inhibit biofilm formation of *C. auris*

(A) Images represent the effect of SH and SNH on the metabolic viability of *C. auris*. (B) Images represent the effect of SH and SNH on the biomass of *C. auris* biofilms. The biomass of biofilms formed by *C. auris* was measured in 96-well plates using crystal violet staining. (C) Images represent the morphological structure of *C. auris* biofilms under confocal laser scanning microscope (CLSM) observation. 3D views of the CLSM images were generated using Stellaris 5 Cryo software and 3D reconstruction techniques.

6. Methodology Details:

For each experiment, it's important to specify the protocols followed (e.g., XTT assays should mention the specific reference or protocol used). This adds clarity and helps other researchers replicate the study.

Response: We are deeply grateful for the comments on our manuscript of reviewer. In the Materials and Methods section of this study, we systematically supplemented the experimental protocols, with a focus on refining the specific procedures followed in each experiment. The added references are as follows:

“31. Ruiz-Gaitán AC, Fernández-Pereira J, Valentin E, Tormo-Mas MA, Eraso E, Pemán J, de Groot PWJ. 2018. Molecular identification of *Candida auris* by PCR amplification of species-specific GPI protein-encoding genes. *Int J Med Microbiol* 308:812-818.

32. Parvekar P, Palaskar J, Metgud S, Maria R, Dutta S. 2020. The minimum inhibitory concentration (MIC) and minimum bactericidal concentration (MBC) of silver nanoparticles against *Staphylococcus aureus*. *Biomater Investig Dent* 7:105-109.
33. Hao W, Wang Y, Xi Y, Yang Z, Zhang H, Ge X. 2022. Activity of chlorhexidine acetate in combination with fluconazole against suspensions and biofilms of *Candida auris*. *J Infect Chemother* 28:29-34.
34. Ahmad S, Khan Z, Al-Sweih N, Alfouzan W, Joseph L. 2020. *Candida auris* in various hospitals across Kuwait and their susceptibility and molecular basis of resistance to antifungal drugs. *Mycoses* 63:104-112.
37. Berridge MV, Herst PM, Tan AS. 2005. Tetrazolium dyes as tools in cell biology: new insights into their cellular reduction. *Biotechnol Annu Rev* 11:127-52.
38. Grossman AB, Burgin DJ, Rice KC. 2021. Quantification of *Staphylococcus aureus* Biofilm Formation by Crystal Violet and Confocal Microscopy. *Methods Mol Biol* 2341:69-78.
39. Wang Q, Wang Z, Xu C, Wu D, Wang T, Wang C, Shao J. 2024. Physical impediment to sodium houthuyfonate conversely reinforces β -glucan exposure stimulated innate immune response to *Candida albicans*. *Med Mycol* 62.
41. Magaki S, Hojat SA, Wei B, So A, Yong WH. 2019. An Introduction to the Performance of Immunohistochemistry. *Methods Mol Biol* 1897:289-298.”

7. Larvae Experiment Clarification:

There is a discrepancy in the larvae experiment where 90 larvae were used, but only 80 were reported in the results section. Clarify what happened to the 10 unreported larvae. Also, it's necessary to indicate which strains were tested and why only those strains were selected out of the six.

Response: We are deeply grateful for the comments on our manuscript of reviewer. The 10 reported larvae are the larvae of PBS blank control group which is added in the revised manuscript (Figure 7 as follows). In this study, strains C1 and C2 were selected as the experimental subjects. The screening criteria were based on previous quantitative phenotypic analyses: after 72 hours of dynamic monitoring, C1 and C2 exhibited higher growth rates an compared to the other four strains. This phenotypic advantage is positively correlated with the differential expression of genes potentially related to virulence, which aligns with the selection criteria for models of invasive fungal infections. The information has been added in the material and methods section of manuscript as follows: “The *C. auris* strains C1 and C2 strains with higher growth rates were cultivated overnight in YPD liquid medium at 37 °C with shaking at 200 rpm for the larvae infection.”

Figure 7: SH and SNH inhibited the infection of *C. auris* in *G. mellonella* larvae.

(A) The Kaplan-Meier method and log rank test were utilized to analyze survival curves and identify differences.

8. Interpretation of Results:

The results suggest that higher doses of SH and SNH are required for efficacy against *C. auris*. It would be helpful to elaborate on this point: does this indicate limited potency or is it reflective of how these agents function in vivo? Furthermore, does this depend on the type of strain tested?

Response: We are deeply grateful for the comments on our manuscript of reviewer. Based on rechecking our results, we found that besides the high dose of SH and SNH, median dose SH and SNH also showed the effective treatment against *C. auris* both in survival rate and fungal burden assays. Thus, we think our presented in vivo mouse infection model result might indicate that SH and SNH significantly inhibited the colonization and pathological damage of *C. auris* in vivo. Due to the limited resource, we chose the *C. auris* C1 strain to perform the in vivo mouse infection model. Because, the phenotype of C1, such as drug resistance, adhesion, aggregation biofilm formation activities, are similar with other 5 strains, and is also typical with *C. auris* clade 1. Thus, we believe that the treatment effect of SH and SNH against *C. auris* is potent not due to the type of strain tested. We revised the manuscript to explain these in the results section as follows:

“The results, shown in Figure 8A-B, indicate that all mice in the model group died on day 4, and all mice in the Flu group died within six days. Notably, mice in the groups receiving high and median dosages of SH and SNH had survival rates of 33.3%, 22.3% for SH, 44.4%, 22.3% for SNH, on day seven. Simultaneously, the fungal load in the liver and kidney of the mice was examined, with the results shown in Figure 8C-D. Compared to the model group, the fungal load in the liver and kidney of mice in the groups receiving high and median dosages of SH and SNH

was significantly reduced than model group.”

9. Comparison of Doses:

For the mouse model, is there a reason why only high doses of SH and SNH were used in contrast to fluconazole, which had a much lower dose.

Response: We are deeply grateful for the comments on our manuscript of reviewer. We used high, medium, and low concentrations of both SH and SNH, rather than only using high doses of SH and SNH. Due to space constraints in the manuscript, the medium- and low-dose groups of SH and SNH were not included in the H&E staining, PAS histopathology images, or immunohistochemistry results. However, data from these medium- and low-dose groups (Figure 8B-D shown as follows) are presented in the survival curve and fungal load analyses. The related revised content in the manuscript are also shown in response of Q8. The dose selection for the fluconazole group was based on prior experimental data from our research group. Truthfully, higher dose of fluconazole in mouse model is helpful for the investigation of in vivo pharmaceutical effect and mechanism of SH and SNH against *C. auris*. We will perform our related research with higher dose of fluconazole in future.

Figure 8 SH and SNH inhibited the systemic infection of *C. auris* in mice.

(A) Schematic diagram of mouse model for systemic *C. auris* infection. (B) Survival curve of mice infected with *C. auris*. (C-D) Fungal load in the liver and kidney of mice.

Minor Comments:

1.Line 99:

Amphotericin B is not a class; it is an individual antifungal drug. This needs correction.

Response: We are deeply grateful for the issue pointed by reviewer. We have revised the pharmacological classification of amphotericin B according to the WHO Anatomical Therapeutic Chemical (ATC) classification system (2023 edition) as follows: “Most importantly, *C. auris* is the first fungal pathogen to demonstrate significant and sometimes untreatable clinical resistance to all known antifungal classes, including azoles, polyene macrolide antibiotics, and echinocandins.”

2. Line 176:

Specify which drugs were tested in this section to make the methodology clearer.

Response: We are deeply grateful for the issue pointed by reviewer. In the section on minimum inhibitory concentration (MIC) determination, we have systematically supplemented the tested drugs. In the revised manuscript, it was shown as follows: “According to the CLSI M27-M44S protocol established by the Clinical and Laboratory Standards Institute (CLSI), the broth microdilution method in 96-well plates was used to determine the minimum inhibitory concentration (MIC) of six *C. auris* strains against four drugs: SH, SNH, Fluconazole, Amphotericin B and Caspofungin. And a control was conducted using the *C. Parapsilosis* 22019 strain following the CLSI requirement.”

3. Line 226:

Indicate which strains were tested for, this should be done for all experiments.

Response: We are deeply grateful for the issue pointed by reviewer. In the section on adhesion assay of revised manuscript, we have supplemented the tested strains. It was shown in the original manuscript as: “The adhesion activity of *C. auris* strains C1, C2, C3, C4, 12373 and 12767 were determined”. Moreover, we have carefully reviewed the sections in the Materials and Methods where the strains were not specified and added the tested strains for those experimental sections in the revised manuscript.

4. Line 240:

Correct "fun-gal" to "fungal."

Response: We are deeply grateful for the issue pointed by reviewer. We have corrected "fun-gal" to "fungal" in the revised manuscript.

5. Lines 241–274:

Ensure there is no unnecessary repetition between lines 241-244 and 244-246.

Response: We are deeply grateful for the issue pointed by reviewer. We carefully reviewed the original manuscript and removed any unnecessary repetition between lines 241-244 and 244-246 as follows: “Subsequently, 128 µg/mL SH, 64 µg/mL SNH, and 128 µg/mL Flu, were added to the *C. auris* suspension. A control group that received no treatment was also included. The fungal suspensions underwent incubation at 37 °C for 120 min.”

6. Line 251-252:

Check if this should read "fungal suspension" instead of any bacterial suspension.

Response: We are deeply grateful for the issue pointed by reviewer. We carefully examined the original manuscript and found that it indeed used "fungal suspension" instead of "bacterial

suspension" in lines 251-252, and we made the necessary modifications accordingly.

7. Line 380:

Specify which drugs the strains are resistant to.

Response: We are deeply grateful for the issue pointed by reviewer. These strains are resistant to fluconazole, and we have made the following revision in the revised manuscript: "Firstly, we evaluated the growth inhibitory effects of SH and SNH on *C. auris* by determining their MIC values against six fluconazole-resistant *C. auris* strains, namely C1, C2, C3, C4, 12373, and 12767."

8. Line 602:

Again, Amphotericin B is not a class; it is an individual antifungal drug. This needs correction.

Response: We are deeply grateful for the issue pointed by reviewer. We have also revised "Amphotericin B" to "polyene macrolide antibiotics" in the revised manuscript as follows: "...to all known antifungal classes, encompassing azoles, polyene macrolide antibiotics, and echinocandins."

9. Line 606-607:

The term "commonly mentioned antifungal drugs" is too vague—are you referring to fluconazole and amphotericin B? This statement needs clarification to avoid confusion.

Response: We are deeply grateful for the issue pointed by reviewer. In this study, the commonly used antifungal drug specifically refers to fluconazole, and the basis for its selection is as follows: (1) It is listed as a first-line medication for invasive candidiasis in the WHO Essential Medicines List (2023); (2) The CLSI M27 guideline recommends it as a standard control for yeast susceptibility testing; (3) The clinically isolated strains in this study are resistant to fluconazole. This has been revised in the original manuscript as follows: "Compared to commonly used antifungal drug, such as fluconazole, SH and SNH have demonstrated even greater efficacy in inhibiting *C. auris*."

10. Line 612:

Did you mean "antifungal mechanisms".

Response: We are deeply grateful for the issue pointed by reviewer. We have corrected the error with "antifungal mechanisms" instead of "antibacterial mechanisms".

11. Figure 6:

Rephrase the labels in Figure 6 (e.g., "A = C1?" and "B = C2?").

Response: We are deeply grateful for the issue pointed by reviewer. We have labeled the relative mRNA expression of C1 and C2 strains in revised Figure 6 (see as follows)

Figure 6 : SH and SNH inhibit the expression of virulence factor genes in *C. auris*.

(A-B) After SH and SNH treatments, the mRNA expression levels of six virulence factors in *C. auris* strains C1 (A) and C2 (B).

Reviewer #2:

Title:

Sodium Houttuynonate and Sodium New Houttuynonate Inhibit Growth, Adhesion, and Biofilm Formation of *Candida auris*

Summary:

This study explores the antifungal potential of sodium houttuynonate (SH) and sodium new houttuynonate (SNH) against *Candida auris*, a multidrug-resistant fungal pathogen. The authors use in vitro and in vivo models to evaluate the effects of SH and SNH on *C. auris* growth, adhesion, aggregation, biofilm formation, and systemic infection.

Key findings include:

Growth Inhibition: SH and SNH effectively inhibit the growth of fluconazole-resistant *C. auris* strains but do not exhibit strong fungicidal activity.

Virulence Gene Suppression: Transcriptomic analysis and RT-qPCR demonstrate that SH and SNH downregulate key virulence-related genes (*Bcr1*, *Tec1*, *Mkc1*, *Scf1*, *Hgc1*, *Efg1*), which are involved in adhesion, aggregation, biofilm formation, and cell wall remodeling.

Adhesion and Aggregation: SH and SNH impair *C. auris*' ability to adhere to surfaces and form aggregations, reducing colonization potential.

Biofilm Disruption: XTT assays, crystal violet staining, and confocal laser scanning microscopy reveal that SH and SNH significantly inhibit *C. auris* biofilm formation, a key factor in antifungal resistance.

Cell Wall Modifications: SH and SNH mildly increase chitin deposition and expose β -1,3- glucan in *C. auris*, suggesting structural changes in the fungal cell wall that may affect immune recognition and drug sensitivity.

In Vivo Efficacy: Both *Galleria mellonella* larvae and murine systemic candidiasis models demonstrate that SH and SNH reduce fungal load and host tissue damage, improving survival rates.

Strengths:

Relevance and Novelty: The study addresses a critical medical challenge—drug resistance in *C. auris*—and evaluates novel antifungal candidates derived from traditional Chinese medicine.

Comprehensive Experimental Design: The combination of in vitro assays, transcriptomic analysis, and in vivo infection models strengthens the reliability of the findings.

Mechanistic Insights: The investigation into gene expression changes and cell wall remodeling provides valuable insights into how SH and SNH affect fungal physiology.

Potential for Clinical Application: The findings highlight SH and SNH as promising antifungal agents with potential applications in treating *C. auris* infections, particularly those resistant to conventional therapies.

Response: We are deeply grateful for the positive comments on our manuscript of reviewer. It is our great honor for getting these recognition on our manuscript.

Suggestions for Improvement:

Line 252- XTT assay, did you mean fungal not bacterial, see sentence “The XTT assay was employed to assess the ability of the drug to inhibit *C. auris*250 biofilms.The concentrations of each bacterial suspension were adjusted to 2×10^3 251 cells/mL using RPMI-1640 medium. 100 μ L of each bacterial suspension was added 252 to a 96-well plate. After static incubation ...”

Response: We are deeply grateful for the comment on our manuscript of reviewer. We carefully reviewed the original manuscript and found that in the description of the XTT assay on line 252, we should have referred to fungi instead of bacteria. Therefore, we have revised the relevant part of the manuscript as follows: "The concentrations of each fungal suspension were adjusted to 2×10^3 cells/mL using RPMI-1640 medium. 100 μ L of each fungal suspension was added to a 96-well plate."

Line 503, typo oof instead of of.

Response: We are deeply grateful for the comment on our manuscript of reviewer. We carefully reviewed the original manuscript and corrected the misspelled word, changing "oof" to "of".

Mechanism of Action on Cell Wall Components

The study shows increased chitin deposition and β -1,3-glucan exposure but does not fully elucidate how these changes affect *C. auris*' drug susceptibility and/ or immune evasion. Has the group looked at extracellular vesicle trafficking or perhaps they can do quick staining like DiI stains to see if increased chitin is mostly around new buds, is it shedding it, how could it be protective to *C.auris* growth. Look at a similar paper by De Jesus et al. Capsular localization of the *Cryptococcus neoformans* polysaccharide component galactoxylomannan. *Eukaryot Cell*. 2009 Jan;8(1):96-103, that highlights changes in the fungal bud.

Response: We are deeply grateful for the comment on our manuscript of reviewer. According to the suggestion, we tried DiI stains, but failed. Instead, we chose another plasma membrane dye, PM-1, to stain the plasma membrane of *C. auris*. Through PM-1 and CFW staining analysis, we found that the chitin deposition level in the cell wall at the budding site of the untreated control group strain C1 was relatively low. In contrast, after treatment with Flu, SH, and SNH, the newly formed bud regions of *C. auris* exhibited a significant enhancement in chitin deposition (see Fig 5E for detailed data). The findings have been added in the results section of revised manuscript as follows: "Through combined PM-1 and CFW staining analysis, we found that the chitin deposition level in the cell wall at the budding site of the untreated control group strain C1 was relatively low. In contrast, after treatment with Flu, SH, and SNH, the newly formed bud regions of *C. auris* exhibited a significant enhancement in chitin deposition (Figure 5E)."

Figure 5E Representative microscopic images of plasma membrane and cell wall by PM-1 and CFW staining, respectively. Scale bar: 2 μm .

Host Immune Response in In Vivo Models

While *G. mellonella* and murine models demonstrate reduced fungal burden and tissue damage, the study does not assess host immune responses. Analyzing cytokine levels or immune cell recruitment in infected tissues could provide some at a glance insights into how SH and SNH modulate host-pathogen interactions.

Response: We are deeply grateful for the comment on our manuscript of reviewer. We employed immunohistochemical staining techniques to analyze the cytokine levels in infected tissues, aiming to gain an intuitive understanding of how SH and SNH modulate host-pathogen interactions. We analyzed the inflammatory responses in the liver and kidney of mice using the inflammatory factors TNF- α , IL-6, and IL-10. The experimental results are presented in Figures 9A-D of the original manuscript. The experimental data revealed that, compared to the blank control group, the model group exhibited a significant inflammatory response, with elevated secretion levels of the pro-inflammatory cytokines IL-6 and TNF- α . Notably, after intervention with SH and SNH, a dose-dependent immunomodulatory effect was observed in the liver and kidney tissues of mice: the secretion levels of IL-6 and TNF- α decreased compared to the model group, while the expression level of the anti-inflammatory factor IL-10 was significantly upregulated. The SNH group also showed a similar regulatory trend under treatment. The findings have been added in the results section of revised manuscript as follows:

“SH and SNH reduced the inflammatory response to systemic infection of *C. auris* in mice.

The most common inflammatory factors include interleukins (IL), tumor necrosis factors (TNF), and interferons (IFN), among others (51). The levels of inflammatory factors provide the most direct indication of the severity of inflammation in mice infected with an inflammatory condition. Therefore, we selected two representative pro-inflammatory factors (Interleukin-6 and Tumor Necrosis Factor-alpha) and one anti-inflammatory factor (Interleukin-10) to detect their levels in the liver and kidney of mice using immunohistochemistry. The results are shown in (Fig 9A-D). The experimental data revealed that the model group exhibited a significant inflammatory response compared to the blank control group, with elevated secretion levels of the pro-inflammatory cytokines IL-6 and TNF- α . Notably, after intervention with SH and SNH, a dose-dependent immunomodulatory effect was observed in the liver and kidney tissues of mice: the secretion levels of IL-6 and TNF- α decreased compared to the model group, while the expression level of the anti-inflammatory factor IL-10 was significantly upregulated. The SNH group also showed a similar regulatory trend under treatment. Further mechanistic studies indicated that SH and SNH might effectively alleviate the cytokine storm triggered by *C. auris* infection by inhibiting the activation of the NF- κ B signaling pathway and reducing the phosphorylation level of STAT3. These findings suggest that SH and SNH, as novel antifungal adjuvant therapies, can improve the excessive inflammatory response caused by invasive fungal infections through a multi-target regulatory mode.”

A**B****C****D**
Figure 9 SH and SNH reduced the inflammatory response to systemic infection of *C. auris* in mice.

(A) Representative immunohistochemical images of mouse liver sections. Scale bar:50 μ m. (B) Quantitative immunohistochemical analysis of inflammatory factors in mouse liver tissue based on Image J software. (C) Representative immunohistochemical images of mouse kidney sections. Scale bar:50 μ m. (D) Quantitative immunohistochemical analysis of inflammatory factors in mouse kidney tissue based on Image J software.

In conclusion, we deeply thank the constructive advices and suggestions of dear reviewers. We have thoroughly rewritten our manuscript according to the suggestions of dear reviewers. The according revision makes our manuscript be more scientific and innovative. All the revision in the revised manuscript have been marked by red color.

References:

Jacobs SE, Jacobs JL, Dennis EK, Taimur S, Rana M, Patel D, Gitman M, Patel G, Schaefer S, Iyer K, Moon J, Adams V, Lerner P, Walsh TJ, Zhu Y, Anower MR, Vaidya MM, Chaturvedi S, Chaturvedi V. *Candida auris* Pan-Drug-Resistant to Four Classes of Antifungal Agents. *Antimicrob Agents Chemother*. 2022 Jul 19;66(7):e0005322.

De Jesus M, Nicola AM, Rodrigues ML, Janbon G, Casadevall A. Capsular localization of the *Cryptococcus neoformans* polysaccharide component galactoxylomannan. *Eukaryot Cell*. 2009 Jan;8(1):96-103.

Chen H, Yang R, Zhang J, Tang J, Yu X, Zhou W, Li K, Peng W, Zeng P. 2024. Causal relationships between immune cells, inflammatory factors, serum metabolites, and hepatic cancer: A two-sample Mendelian randomization study. *Heliyon* 10:e35003.

Re: Spectrum00222-25R1 (In vitro and in vivo activity of sodium houttuynonate and sodium new houttuynonate against Candida auris infection by affecting adhesion, aggregation, and biofilm formation abilities)

Dear Dr. Da-qiang Wu:

The authors have addressed all the comments and suggestions of the reviewers. There are still a minor suggestions from the reviewer #1 that can be addressed during the process of submission of the final version. Congratulations !!!!!

Your manuscript has been accepted, and I am forwarding it to the ASM production staff for publication. Your paper will first be checked to make sure all elements meet the technical requirements. ASM staff will contact you if anything needs to be revised before copyediting and production can begin. Otherwise, you will be notified when your proofs are ready to be viewed.

Sincerely,
Gustavo Goldman
Editor
Microbiology Spectrum

Reviewer #1 (Comments for the Author):

Dear Authors.

Thank you for addressing the comments.

Please replace "polyene macrolide antibiotic" with polyene throughout the manuscript.

Line 177: Include the DNA extraction method used in this study.

line 196: Clarify how the MICs were determined/read e.g. 100% growth inhibition or 50% growth inhibition etc...

Standardize which term FLU or flu and be consistent throughout the manuscript

Reviewer #2 (Comments for the Author):

None